# Hippocampal cells integrate past memory and present perception for the future

Cen Yang[1,2,3], Yuji Naya[1,3,4,5]*

**1** School of Psychological and Cognitive Sciences, Peking University, Beijing, China, **2** Academy for Advanced Interdisciplinary Studies, Peking University, Beijing, China, **3** Center for Life Sciences, Peking University, Beijing, China, **4** PKU-IDG/McGovern Institute for Brain Research, Peking University, Beijing, China, **5** Beijing Key Laboratory of Behavior and Mental Health, Peking University, Beijing, China

* yujin@pku.edu.cn

**Data Availability Statement:** The source data for all figures in this manuscript are available as supporting information in S1 Data and S2 Data. The original data of spike and eye position are available at the following osf.io repository: https://

## Abstract

The ability to use stored information in a highly flexible manner is a defining feature of the declarative memory system. However, the neuronal mechanisms underlying this flexibility are poorly understood. To address this question, we recorded single-unit activity from the hippocampus of 2 nonhuman primates performing a newly devised task requiring the monkeys to retrieve long-term item-location association memory and then use it flexibly in different circumstances. We found that hippocampal neurons signaled both mnemonic information representing the retrieved location and perceptual information representing the external circumstance. The 2 signals were combined at a single-neuron level to construct goal-directed information by 3 sequentially occurring neuronal operations (e.g., convergence, transference, and targeting) in the hippocampus. Thus, flexible use of knowledge may be supported by the hippocampal constructive process linking memory and perception, which may fit the mnemonic information into the current situation to present manageable information for a subsequent action.

## Introduction

Declarative memory enables individuals to remember past experiences or knowledge and to use that information according to a current situation [1, 2]. This flexible use of stored information is in contrast to procedural or fear-conditioned memory, in which acquired memory is expressed in a fixed form of associated actions or physiological responses [3–5]. Previous studies revealed the involvement of the hippocampus (HPC) in the medial temporal lobe (MTL) in the formation and retrieval of declarative memory [2, 3, 6–14]. However, the mechanism by which the HPC contributes to the flexibility in the usage of the declarative memory remains largely unknown.

The contribution of the HPC to declarative memory was often investigated by examining its spatial aspects in both human subjects [15–17] and animal models [3, 8–10, 13, 14, 18–20]. In the preceding literature, the contributions of the HPC to the spatial memory task were successfully dissociated from those of the other brain areas when the start position in spatial mazes differed between training (e.g., "south" in a plus maze) and testing trials (e.g., "north")

osf.io/nu9ch/?view_only=
1faa4cc2d5254b6eb25740a92e6f693c.

**Funding:** This work was supported by National
Natural Science Foundation of China (http://www.
nsfc.gov.cn/english/site_1/index.html; grant
numbers 31471076 and 31421003 to YN). The
funder had no role in study design, data collection
and analysis, decision to publish, or preparation of
the manuscript.

**Competing interests:** The authors have declared
that no competing interests exist.

**Abbreviations:** AUC, area under curve; CMP,
constructive memory-perception; fMRI, functional
magnetic resonance imaging; HPC, hippocampus;
MTL, medial temporal lobe; PFC, prefrontal cortex;
PRC, perirhinal cortex; ROC, receiver operating
characteristic; SDF, spike density function.

[3, 21], because the fixed action patterns acquired during the training period (e.g., "turn left," egocentric coordinate) cannot guide the subjects to a goal position (e.g., "west," allocentric coordinate) in the testing trials. The HPC thus contributes to the memory task by retrieving a goal position, which could be represented in an acquired cognitive map [22]. However, in order to reach the goal position, it is not enough for the subjects to remember the goal on the cognitive map, which represents the allocentric spatial relationship of the environment in mind. In addition, it would be critical for the subjects to locate their self-positions by perceiving current circumstances around them and relate the goal to the self-positions in egocentric coordinate for a subsequent action. The subjects thus need to transform the goal position within the cognitive map into goal-directed information relative to a specific circumstance (i.e., start position) that the subjects currently experience. In the present study, we hypothesized that the goal-directed information in the specific circumstance may be constructed by combining the retrieved memory and incoming perceptual information on the same principle as "constructive episodic memory system" suggested by human neuroimaging studies [1, 23, 24]. In this theory, the constructive episodic memory system recombines distributed memory elements for both remembering the past and imagining the future (i.e., "mental time travel") [25].

We therefore investigated whether and how the HPC neurons combine the retrieved location with the perceived circumstance in order to construct goal-directed information. To achieve this purpose, we devised a new memory task for macaque monkeys, in which memory retrieval and its usage were separated by sequential presentations of 2 cues in a single trial (Fig 1). The first cue presented a visual item (item cue) that would trigger retrieval of the location associated with the item. The second cue presented a background image (background cue) that would be combined with the retrieved location to construct goal-directed information. This task structure allowed us to separate the constructive process from the retrieval of item-location association memory. In addition, the animals were prompted to link the individual items to the preassigned locations on the background image through repetitive trainings. Taken together, we investigated the constructive process to fit the semantic-like memory (cf., episodic memory) to the current situation in the present task. We referred to this new task as the constructive memory-perception (CMP) task.

By measuring single-unit activities during the CMP task, we examined whether and how the retrieved memory and incoming perceptual signals were combined in the HPC. One hypothesis might be that the 2 signals would be directly linked to the goal-directed information by a conjunctive representation [20, 26, 27], which binds input elements into a unitary representation and supports the "hippocampus indexing theory" [45]. An alternative hypothesis might be that the memory and perceptual signals converge on the responses of single HPC neurons holding both signal contents [20]. This "convergence" process would require an additional neuronal operation to transfer the retrieved location to the target ("transference" process) and then to represent the target location itself ("targeting" process), which would be analogous to the conjunctive representation.

The present study supported the sequentially occurring neuronal operations in the HPC consisting of the "convergence," "transference," and "targeting" processes. The HPC may equip the declarative memory with flexibility in its usage by the constructive process combining memory and perception through the 3 neuronal operations.

## Results

### CMP task

Two rhesus macaques were trained to perform the CMP task. In the CMP task, 4 pairs of visual items were assigned to 4 different locations (co-locations) on a background image (Fig 1A and

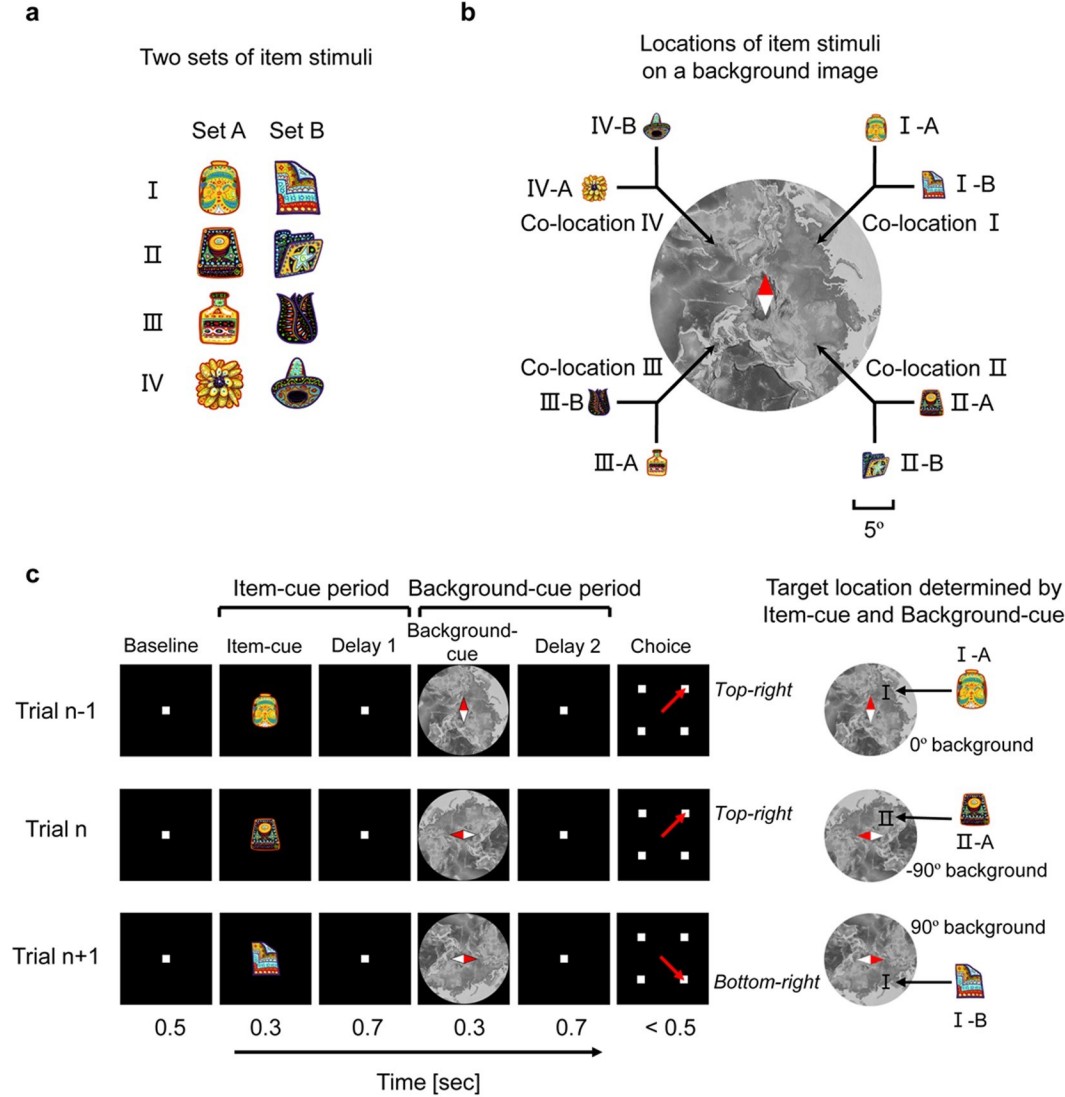

**Fig 1. CMP task. (a)** Item stimuli. **(b)** Item-location association pattern. Two items, one from set A (e.g., I-A) and the other from set B (e.g., I-B), were assigned to each location (e.g., co-location I) on the background image. Scale bar for both item-cue and background-cue stimuli, 5˚ visual angle. **(c)** Schematic diagram of the CMP task. An item cue and background cue were chosen pseudorandomly in each trial. The monkeys should maintain fixation on the center until the end of the background-cue period including Delay 2, then saccade to the target location (red arrow) during the choice period. Monkeys were trained using every 0.1˚ step in orientation from −90˚ to 90˚, though only 5 orientations (−90˚, −45˚, 0˚, 45˚, and 90˚) were tested during the data acquisition. Relative sizes of the item-cue stimuli to the background-cue stimuli were magnified for display purpose. CMP, constructive memory-perception.

1B, S1 Text), which would be stored as long-term association memory linking the items and their locations in allocentric coordinate. We referred to the 2 items in each pair as "co-location" items (e.g., I-A and I-B) because the 2 items were assigned to the same location on the background image. The configuration of the co-location items allowed us to evaluate an item-location memory effect for each single neuron by examining the correlation in its responses to the co-location items. In the present study, we used the same 8 visual items and 1 background image during all the recording sessions. In each trial, 1 of the 8 items was presented as an item cue (e.g., II-A) (Fig 1C). After a short delay, a randomly oriented background image was

presented as a background cue (e.g., −90°). The subjects were then required to saccade to the target location (e.g., top-right), which would be represented in egocentric coordinate, determined by the combination of the item and background cues (e.g., co-location II on the −90°-oriented background on the display).

In the initial training, the monkeys learned the item-location association through trials with a fixed orientation of background cue (which we defined as 0°). After they learned the association between items and locations in trials with the 0° background cue, orientation of the background cue was randomly chosen from −90° to 90° (in 0.1° steps). During the recording session, the orientation of the background cue was pseudorandomly chosen from among 5 orientations (−90°, −45°, 0°, 45°, and 90°). The 2 monkeys performed the task correctly (chance level = 25%) at rates of 80.9% ± 8.1% (mean ± standard deviation; monkey B, $n = 179$ recording sessions) and 96.8% ± 3.1% (monkey C, $n = 158$ recording sessions). Neither of the animals showed any strong bias in the performance among item-cue identities, background-cue orientations, or target locations (S1 Fig). While the monkeys performed the task, we recorded single-unit activity from 456 neurons ($n = 247$ for monkey B, $n = 209$ for monkey C) in the HPC of the MTL (S2 Fig, S1 Table).

## Representation of the retrieved memory

We first investigated the retrieval process during the item-cue period of the task. Fig 2A shows an example of a neuron exhibiting item-selective activity (item-selective neuron, $P < 0.01$, 1-way ANOVA). This neuron exhibited the strongest response to item I-A (optimal), whereas an item paired with the optimal item (I-B, pair) elicited the second-strongest response from the same neuron. The neuron thus strongly responded to only the particular co-location items (i.e., I-A and I-B) but not to others (Fig 2B). The selective responses to I-A and I-B could not be explained by eye position (S3 Fig). To examine the item-location association effect, we calculated the Pearson correlation coefficient between the responses to the 4 pairs of co-location items and referred to it as the co-location index (S4 Fig). Therefore, the co-location index was influenced by the responses not only to the optimal and its paired co-location items but also all other items. If a single neuron in a population showed the pattern of stimulus selectivity that was independent of the items' co-locations, the mean value of the co-location index for the neuronal population would be expected to approach zero as the number of neurons in the population increased. The co-location index of this neuron was extremely high (Fig 2B) ($r = 0.99$, $P < 0.0001$, permutation test, 2-tailed), which indicates a strong long-term memory effect on the responses of this neuron.

Fig 2C shows the population-averaged spike density functions (SDFs) of item-selective neurons ($n = 136$) to their optimal items, paired items, and other items (average across 6 items). The responses to the items paired with the optimal items were significantly larger than those to the other items during the item-cue period ($P < 0.01$ for each time step, $t$-test, 2-tailed). The item-selective neurons also showed extremely large co-location index values ($r = 0.89$, median) (Fig 2D). We confirmed that the large co-location index values could not be explained by eye position (S2 Text). These results indicated that the HPC showed an item-location association effect on the item-selective activities.

The item-location association effect, revealed by the co-location index using the Pearson correlation coefficient, suggests 2 possible response patterns in the HPC during the item-cue period: the neuronal responses representing the locations retrieved from the item cues and those representing individual items that were modulated by the co-locations. If the former holds true, the neurons would not distinguish the co-location items because they would signal the same location. Conversely, if the latter holds true, the neurons would discriminate between

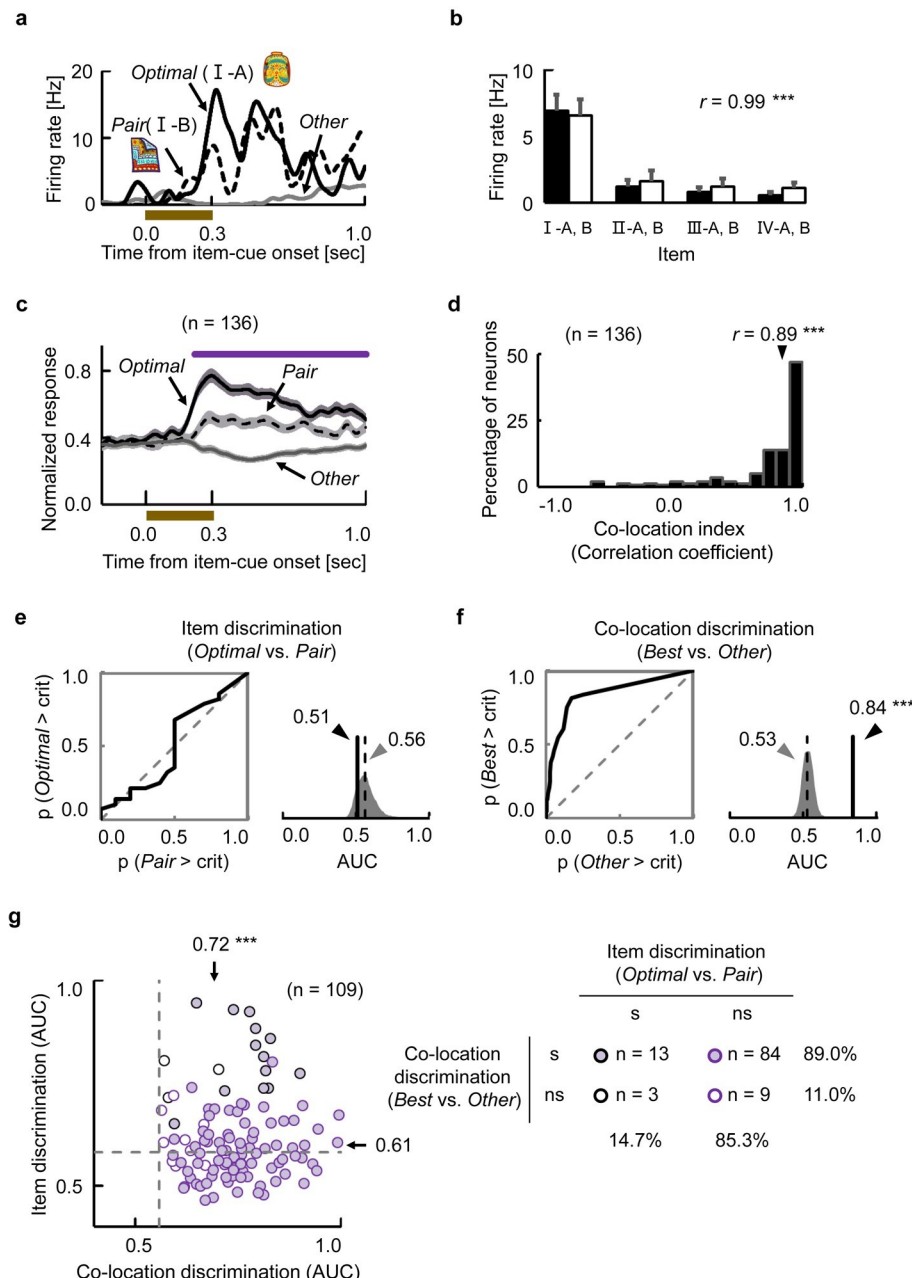

**Fig 2. Representation of the retrieved memory during the item-cue period. (a)** Example of an item-selective neuron with the co-location effect. Black lines indicate SDFs in trials with the Optimal and its Pair items (i.e., best co-location items) of the neuron. Gray line indicates averaged response in the other trials (Other). Brown bar, presentation of the item cue. **(b)** Mean discharge rates and SEM of the same neuron during the item-cue period for each item. Black bars, set A. White bars, set B. *r*, co-location index. ***$P < 0.0001$, permutation test, 2-tailed. **(c)** Population-averaged response of item-selective neurons (*n* = 136). SDFs in trials with the best co-location items (i.e., Optimal and Pair) and other items. Shading, SEM. Purple line, time duration indicating a significant ($P < 0.01$, *t*-test, 2-tailed) difference between pair and other. **(d)** Distributions of co-location indices for item-selective neurons (*n* = 136). *r*, median value. ***$P < 0.0001$, Wilcoxon's signed-rank test, 2-tailed. **(e)** Response discriminability between the optimal and its paired items of the same example neuron in Fig 2A. (Left) ROC curve. (Right) Solid vertical line, AUC value of the example neuron. Gray area and dashed vertical line, distribution of simulated AUC values and its median. **(f)** Response discriminability between best and other co-locations of the same neuron. ***$P < 0.0001$, permutation test, 1-tailed. **(g)** Two-dimensional **s**catter plots of AUC values between the item (ordinate) and co-location (abscissa) discriminations for item-selective neurons with high co-location index ($r > 0.6$; *n* = 109). Each circle indicates one neuron. Arrow,

median of real AUC values for each discrimination. Dashed line, median of simulation AUC values for each discrimination. ***$P < 0.0001$, Wilcoxon's signed-rank test, 2-tailed. s, significant, $P < 0.05$ for each neuron, permutation test, one-tailed. Source data are available in S1 Data. AUC, area under curve; ns, nonsignificant; ROC, receiver operating characteristic; SDF, spike density function.

the co-location items, although the responses to the co-location items were correlated. To test these alternative assumptions, we conducted the receiver operating characteristic (ROC) analysis for the item-selective neurons that showed disproportionately high co-location index ($r > 0.6$, 80% of the recorded neurons) (Fig 2D). We calculated the corresponding area under curve (AUC) for each neuron and examined whether the value was significantly ($P < 0.05$) larger than the chance level, which was estimated by a permutation test (see Methods). Fig 2E indicated response discriminability between the optimal and its paired items of the same neuron in Fig 2A. The ROC curve was close to the diagonal line from (0, 0) to (1, 1), and its AUC value (0.51) was even smaller than an expected value (median AUC = 0.56 in 10,000 permutations). Conversely, the same neuron showed significant response discriminability between best and other co-locations (AUC = 0.84, $P < 0.0001$, permutation test, 1-tailed) (Fig 2F). Out of the 109 neurons with the high co-location index, 93 neurons (85%) could not discriminate optimal items from their paired items (Fig 2G) even with the use of a liberal threshold of statistical significance ($P < 0.05$, one-tailed). We confirmed that 89% of the 109 neurons successfully discriminated the best co-location items, including the optimal items and their paired items (e.g., I-A and I-B for the neuron in Fig 2A), from other co-location items. These results indicate that the HPC neurons exhibited "unitized" [28, 29] responses to the co-location items, implying activation of the same location (i.e., co-location) information to be retrieved from the co-location items.

## Retrieval signal after background-cue

We examined item-selective activity during the background-cue period using a 3-way ANOVA with item cue, background cue, and target position effects as main factors for each neuron ($P < 0.01$) and found a substantial number of item-selective neurons (47 out of 456 neurons) (Fig 3A and 3B). The item-selective neurons during the background-cue period showed larger co-location index values (median $r = 0.94$, $P < 0.01$, Kolmogorov–Smirnov test) (Fig 3C) than those during the item-cue period. Moreover, 44 out of the 47 item-selective neurons (94%) could not distinguish individual items of the best co-locations significantly ($P < 0.05$, permutation test, 1-tailed) (Fig 3D). These results suggested a strong unitization effect on the item-selective activities in each co-location during the background-cue period. Therefore, we designed a 3-way nested ANOVA in which individual co-location items were under their co-locations to examine the main effects of the "co-location," "background," and "target" on neuronal responses during the background-cue period for each neuron (S1 Table, S3 Text). The 3-way nested ANOVA showed that 66 out of the 456 recorded neurons exhibited significant ($P < 0.01$) co-location effects on their activities during the background-cue period. Out of them, 30 neurons exhibited the co-location-selective activities only after the background-cue presentation (Fig 3A and 3B, S5A Fig), which might be recruited to signal the retrieved location in the HPC for the necessity of the constructive process during the background-cue period (S5B Fig).

## Convergence of the retrieved memory and incoming perception

We next investigated how the incoming background-cue information affected the retrieved location signal. Fig 4A shows an example of a neuron exhibiting selective responses to the

**a**

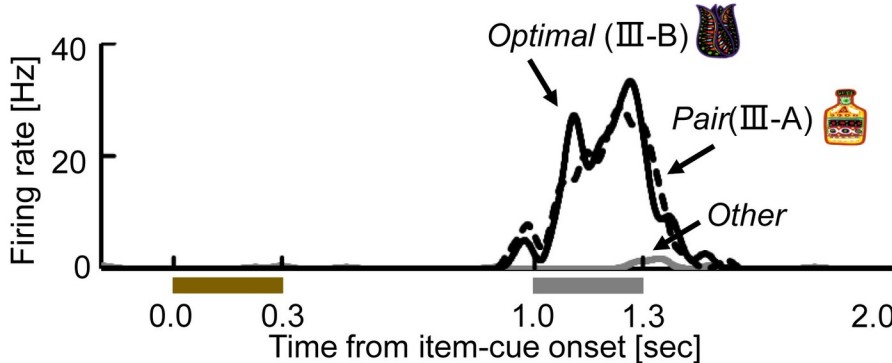

**b**

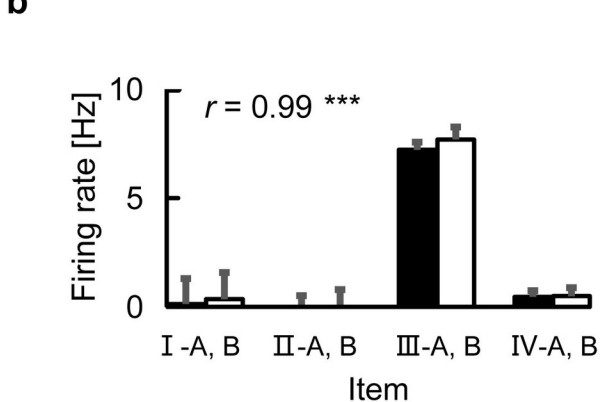

**c**

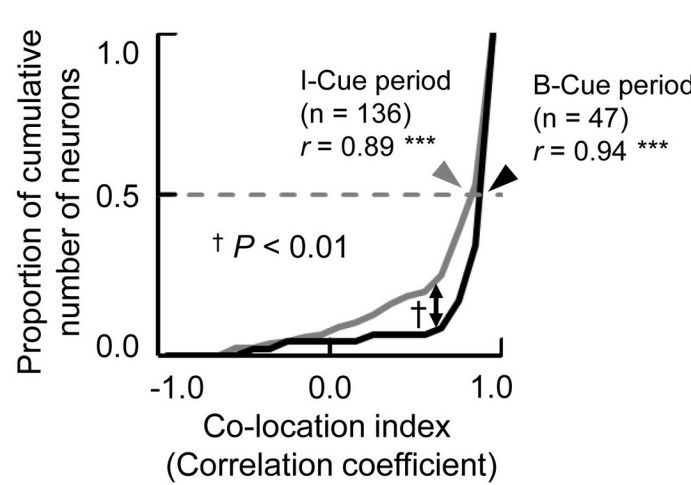

**d**

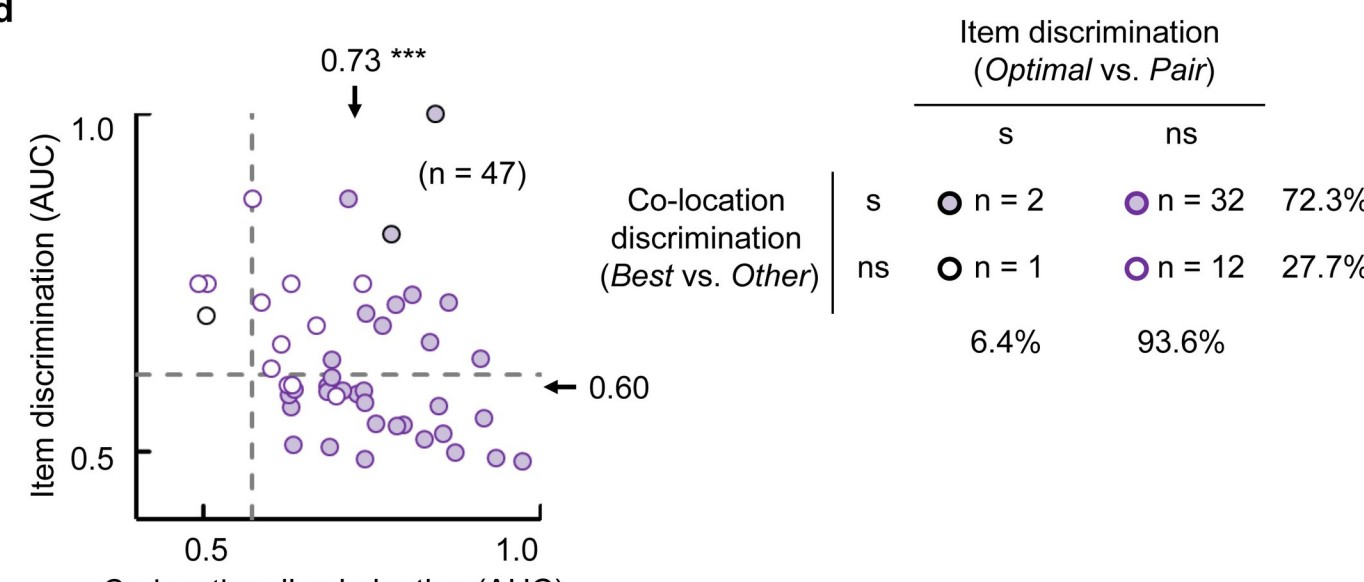

**Fig 3. Retrieval signal after background-cue.** (a) Example of an item-selective neuron with co-location effect. Black lines indicate SDFs in trials with the Optimal and its Pair items (i.e., best co-location items) of the neuron. Gray line indicates averaged response in the other trials (Other). Brown bar, presentation of the item-cue. Gray bar, presentation of the background cue. (b) Mean discharge rate and SEM of the same neuron during the background-cue period for each item. Black bars, set A. White bars, set B. $r$, co-location index. ***$P < 0.0001$, permutation test, 2-tailed. (c) Cumulative frequency histograms of the co-location index. Black line, item-selective neurons during B-Cue period. Gray line, item-selective neurons during I-Cue period. $r$, median of the co-location index values. ***$P < 0.0001$, Wilcoxon's signed-rank test, 2-tailed. †$P < 0.01$, Kolmogorov–Smirnov test. (d) Two-dimensional scatter plot of AUC values for item-selective neurons during the background-cue period ($n = 47$). Same format as Fig 2G. ***$P < 0.0001$, Wilcoxon's signed-rank test, 2-tailed. s, significant, $P < 0.05$, permutation test, 1-tailed. Source data are available in S1 Data. AUC, area under curve; B-Cue, background cue; I-Cue, item cue; ns, nonsignificant; SDF, spike density function.

background cues ($P < 0.01$, 3-way nested ANOVA). This neuron showed the strong responses across all the co-locations when the orientation of the background cue was either 90˚ or 0˚, whereas it showed only negligible responses when the orientation was −90˚. An amplitude of the background-cue effect was exhibited as a time course of the $F$ value (gray curve, middle panel), which characterized the transient increase of the background-cue effect on the neuron's responses regardless of the co-locations of item-cues (yellow curve) and target positions (black curve). In addition to the neurons showing only background-selective activity (e.g., Fig 4A), we found neurons showing selectivity for both co-locations and backgrounds. An example neuron in Fig 4B began signaling co-locations III and IV at the end of the item-cue period. After the background-cue presentation, this neuron exhibited additional excitatory responses for the best co-locations (i.e., III and IV), especially when the orientation of background cue was 90˚. The background-selective responses were thus combined with the co-location-selective responses in this individual neuron (see also S6A Fig), which was shown by the overlap between the co-location (yellow curve) and background (gray curve) effects indicated by their $F$ values (middle panel). We further evaluated the similarity of orientation tuning across co-locations for the example neuron by calculating the Pearson correlation coefficient between the responses to the different orientations of the background-cues for the "best co-location" (III) and those for the "second-best co-location" (IV) (Fig 4B). We found high similarity of orientation tuning across the co-locations ($r = 0.95$). These results indicate that this neuron signaled the background cue irrespective of which co-location signal the neuron held from the item-cue period.

After background-cue presentation, a substantial number of neurons (22% of the recorded neurons, $P < 0.01$, 3-way nested ANOVA) exhibited either co-location-selective activities (14%, 66 neurons) or background-selective activities (14%, 66 neurons). Importantly, a significantly larger number of neurons ($n = 32$, $P < 0.0005$, $\chi^2$-test) showed both co-location and background-cue effects on their activities than the expected number (i.e., 66/456 × 66/456 × 456 = 9.6) (S1 Table). We further evaluated the background-cue effect on the co-location-selective activities in the HPC by examining the similarity of orientation tuning for each of the co-location-selective neurons during the background-cue period ($n = 66$). To do this, we calculated the correlation coefficient at each instantaneous time point (100 milliseconds of time-bin) after the background-cue onset. Here, a positive value of the correlation coefficient would imply a similar orientation tuning across co-locations. A similarity between the orientation tunings was observed from 228 to 458 milliseconds after the background-cue onset in the population ($P < 0.01$ for each time step, Wilcoxon's signed-rank test) (Fig 4C). These results suggested a convergence of the retrieved location and perceptual information on the single-neurons, which transiently held both signal properties on their activity ("convergent-type"), rather than a conjunctive representation.

## Representation of retrieved location and target location

After the background-cue presentation, some co-location-selective neurons exhibited target location selectivity. For example, a neuron in Fig 5A responded to item cues that were assigned

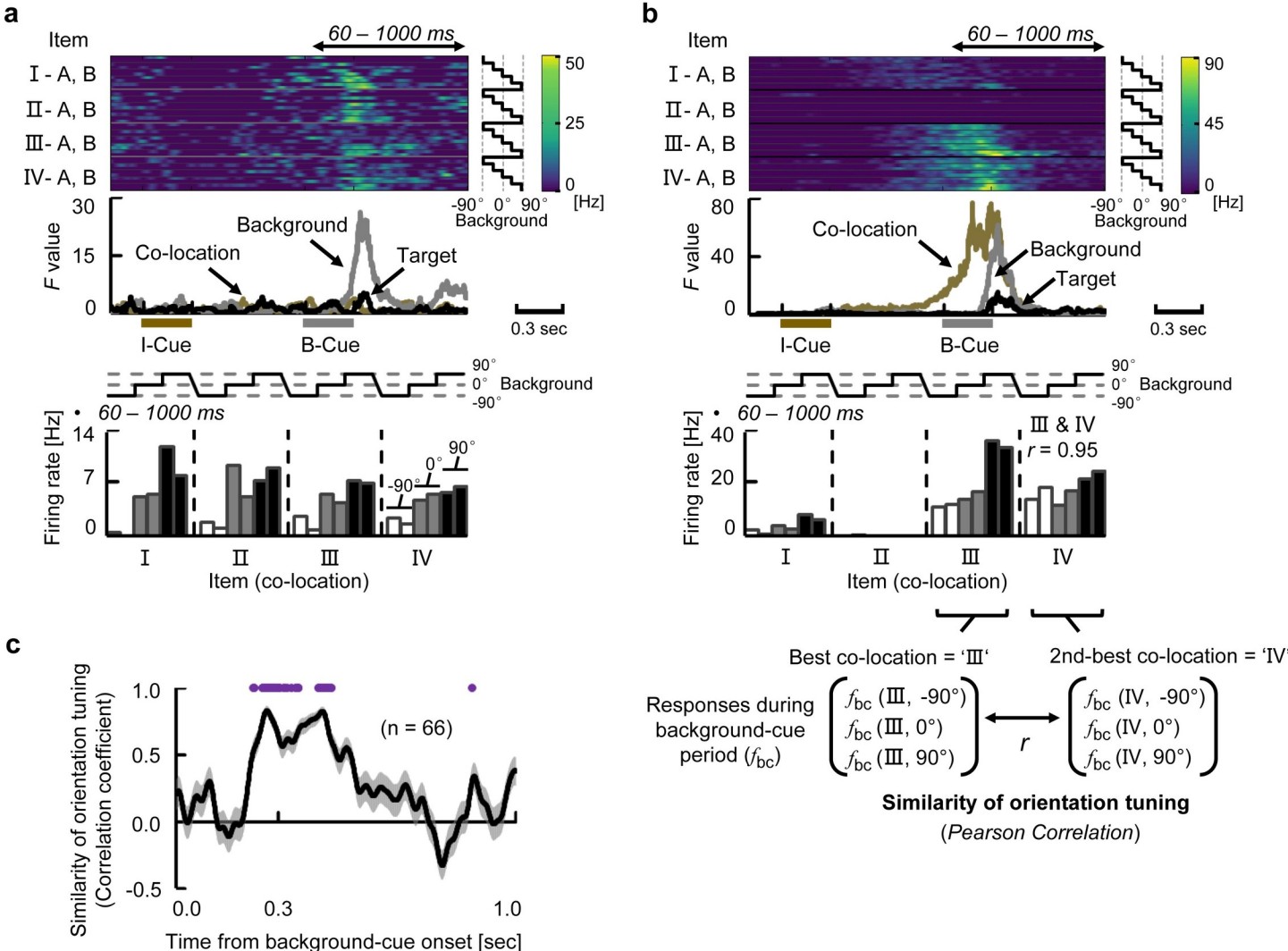

**Fig 4. Convergence of the retrieved memory and incoming perception. (a)** Example neuron signaling background effect. (Top) Each row contains an SDF for each combination of I- and B-Cues. (Middle) Time courses of *F* values. Brown bar, presentation of the I-Cue. Gray bar, presentation of the B-Cue. (Bottom) Mean discharge rate for each combination of I- and B-Cues during 60–1,000 milliseconds after B-Cue onset. White, gray, and black bars indicate −90˚, 0˚, and 90˚ orientations of background cue, respectively. Two bars with the same grayscale indicate the co-location items (e.g., left bar, I-A; right bar, I-B). **(b)** Example neuron signaling background and co-location effects in a "convergent" manner. Same format as Fig 4A. *r*, Pearson correlation coefficient between the orientation tunings (responses to −90˚, 0˚, and 90˚ during the B-Cue period) for "best co-location" (III) and for "second-best co-location" (IV). **(c)** Time course of similarity of orientation tuning *r*(t). Line and shading, means and SEMs of the similarity of the orientation tunings for co-location-selective neurons. Purple line, time duration in which the similarity was significantly positive ($P < 0.01$, $n = 66$, Wilcoxon's signed-rank test for each time step, 2-tailed). Source data are available in S1 Data. B-Cue, background cue; I-Cue, item cue; SDF, spike density function.

to the co-location III during the item-cue period, whereas the same neuron showed selective activity for a particular target location that corresponded to the bottom-left (yellow) during the background-cue period. The bottom-left of the target location matched to the co-location III if we assume the background image with 0˚ orientation. The responses to the target locations during the background-cue period were largely correlated with those to the co-locations during the item-cue period when the co-locations were assumedly positioned relative to the 0˚ background image (matching index, $r = 0.99$) but not to the −90˚ ($r = −0.28$) nor the 90˚ ($r = −0.23$) background image (Fig 5B). This result may imply that the item-location is retrieved

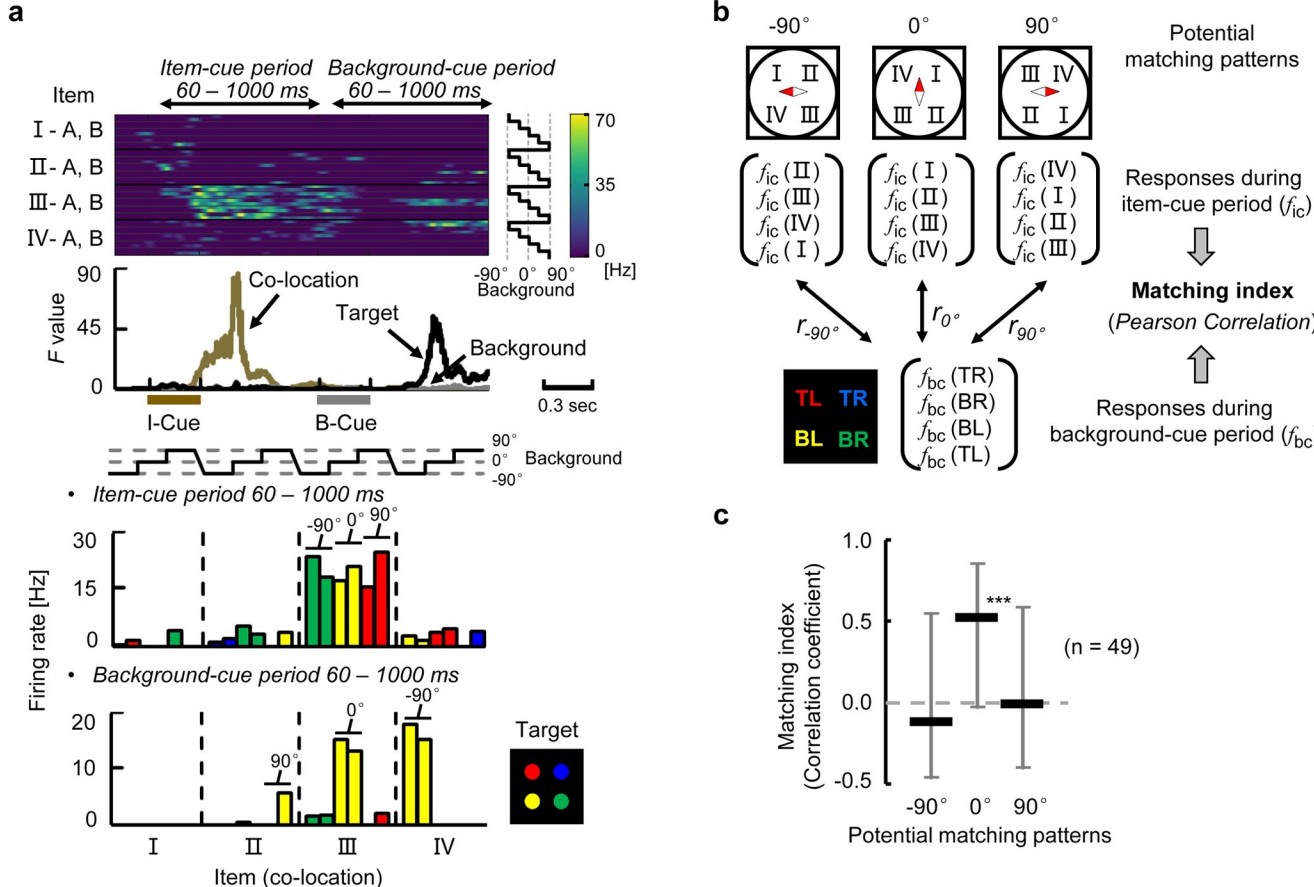

**Fig 5. Representation of retrieved location and target location. (a)** Example neuron signaling a particular co-location during the item-cue period and a particular "*targeting*" location during the B-Cue period. Same format as Fig 4A, except that target locations in the bar graph are indicated by colors (bottom panel). For example, yellow color corresponds to the bottom-left of the target location on the display. **(b)** Potential matching patterns between the co-location and target location. **(c)** Median value of matching index for each matching pattern (using neurons signaling both co-location-selectivity during the item-cue period and target-selectivity during the background-cue period, $n = 49$). Error bar, quarter value. ***$P < 0.0001$, Wilcoxon signed-rank test, 2-tailed. Source data are available in S1 Data. B-Cue, background cue; BL, bottom left; BR, bottom right; I-IV, co-location I-IV; I-Cue, item cue; TL, top left; TR, top right.

relative to the 0˚ background image as default. The presence of the default position/orientation of the background image in a mental space of the monkeys may reflect the effect of initial training, during which the monkeys learned the combinations of items and locations in trials with the 0˚ background cue. To test this implication, we collected 49 neurons showing significant target-selectivity out of the 136 neurons with item-selectivity during the item-cue period. These neurons tended to show the preferred target locations that corresponded to the preferred co-locations relative to the 0˚ background cue (default orientation) but not to the other orientations during the item-cue period (Fig 5C). It should be noted that if these neurons represented the retrieved location relative to the background image only in allocentric coordinate without projecting it into egocentric coordinates (first person's perspective), their preferred co-locations and target locations would be independent, and a population average of the correlation coefficients (matching index) would be close to zero value in any orientation of the background cue. The presence of the default position/orientation of the background image implies that the HPC might represent the retrieved location in the egocentric space (first person's perspective) rather than the allocentric space.

## Construction of the goal-directed information

We next investigated how the retrieved location was transformed to the target location when the background cue was presented. Fig 6A shows an example neuron that exhibited strong transient responses to particular combinations of item cues and background cues ([I, 90˚] and [III, −90˚]) corresponding to the same target location (bottom right, green). However, this neuron did not respond when the background cue was 0˚ even though the combination (II,

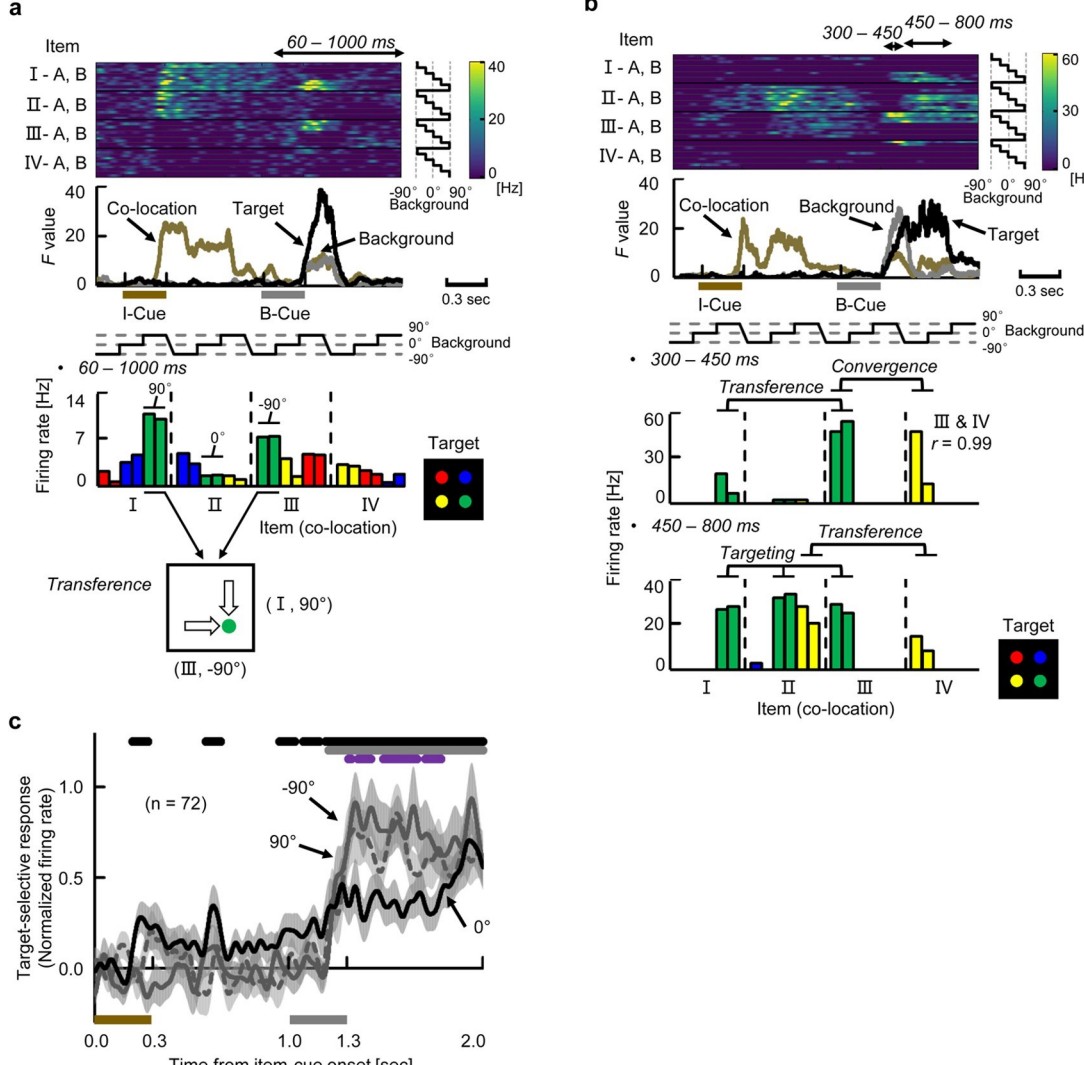

**Fig 6. Construction of goal-directed information. (a)** Example neuron showing the "transference" effect ([I, 90˚] and [III, −90˚] for the bottom-right). Same format as Fig 5A. Bottom panel shows a schematic diagram of "transference" from the retrieved location (co-locations I and III) into the same target location (bottom right, green). **(b)** Example neuron showing multiple operations. **(c)** Target-selective responses ("best" minus "others") in trials with −90˚, 0˚, and 90˚ background cues for target-selective neurons (n = 72). Curves and shadings depict means and SEMs of population-averaged SDFs. Top lines, time duration in which target-selective responses ("best" minus "others") were significantly positive in trials with 0˚ background cues (black) and with either −90˚ or 90˚ background cues (gray) (P < 0.05, t test, 2-tailed). Purple line, time duration in which the "best" target responses were significantly larger in trials with either −90˚ or 90˚ compared with 0˚ background cue (P < 0.05). The best target locations of the target-selective neurons in each hemisphere (animal) covered not only the contra-lateral side but also ipsi-lateral side (S2 Table). Source data are available in S1 Data. B-Cue, background cue; I-Cue, item cue; SDF, spike density function.

0˚) corresponded to the same target location (bottom right). This implies that the neuron responded only when the retrieved location was transferred to the preferred target location of the neuron (i.e., bottom right). The target signal of this example neuron thus depended on the preceding 2 cues, which was demonstrated by the increased $F$ values for all 3 main effects. This "transferring-type" of activity contrasts with the target-selective activity of the neuron shown in Fig 5A, which signaled the preferred target location itself regardless of the co-locations of item cues or the orientations of background cues. We refer to the latter type of target-selective activity as "targeting-type," which was characterized by a robust increase of the $F$ values only for the target effect (black curve, Fig 5A, S6B Fig). Interestingly, some individual neurons exhibited convergent-type activity first, then transferring-type activity, and finally targeting-type activity (Fig 6B, S6C Fig). These results imply a temporal relationship between the transference effect and targeting effect during the construction of goal-directed information.

To examine the temporal relationship at the population level, we compared time courses of the 2 types of target-related effects for the target-selective neurons ($n = 72$) by examining the effect of background-cues in different orientations (−90˚, 0˚, and 90˚) on the target-selective responses (Fig 6C). The target-selective responses in trials with the −90˚ and 90˚ background-cues became significantly larger than those with the 0˚ background cue from 309 to 786 milliseconds after the background-cue onset (Fig 6C). The increase in target-selective responses after the −90˚ and 90˚ background cues may reflect the transfer of the retrieved location into the preferred locations of individual HPC target-selective neurons ("transferring-type"). Then, the target-selective responses in trials with the 0˚ background-cue began to increase in the middle of delay 2, and the target-selective responses ultimately became indistinguishable among all the background-cues (Fig 6C), which may represent the target locations themselves ("targeting-type"). In trials with a 0˚ background cue, target-selective responses were observed not only during the background-cue period but also during the item-cue period ($P < 0.05$, $t$ test, 2-tailed) (Fig 6C), which confirmed the presence of the default position/orientation of the background image for the representation of the retrieved item-location in the HPC. Considering the fact that the immediate background-cue effect converged on the retrieved location signal, these results suggest involvements of sequentially occurring neuronal operations (convergence, transference, and targeting) in the constructive process in which both memory and perception were combined to generate a goal-directed representation of the memory (S1 Table).

## Neuronal signal predicts animals' behavior

We finally investigated whether the target-selective responses in the HPC were correlated with subjects' behaviors. For this purpose, we conducted an error analysis for the target-selective activities during the background-cue period. Fig 7A shows an example neuron exhibiting target-selective activities. This neuron showed strong responses during the background-cue period when the animal chose the top-left position (red) not only in the correct trials (Correct trials, red) but also in the error trials (False Alarm, black). In contrast, the neuron did not respond when the animal made mistakes by missing the top-left target position (Miss, gray). We examined whether the target-selective activities in the error trials could be explained by the positions the animals chose or the correct positions of the trials using partial correlation coefficients (see Materials and methods). The activities in the error trials of this neuron were related with the animals' choice ($r = 0.51$, $P < 0.0001$, $d.f. = 47$) but not with the correct position ($r = −0.18$, $P = 0.94$). We calculated the partial correlation coefficients for the target-selective neurons with more than 10 error trials and found that the activities in error trials reflected the animals' choice rather than the correct position ($P < 0.0005$, Wilcoxon's signed-rank test)

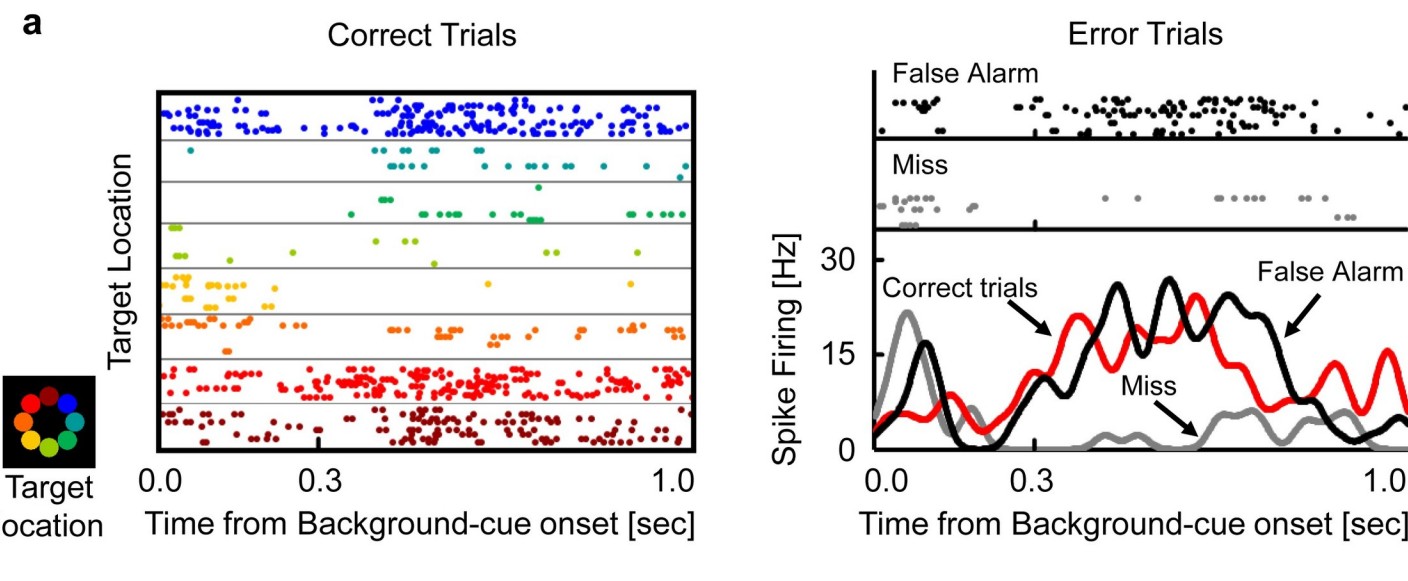

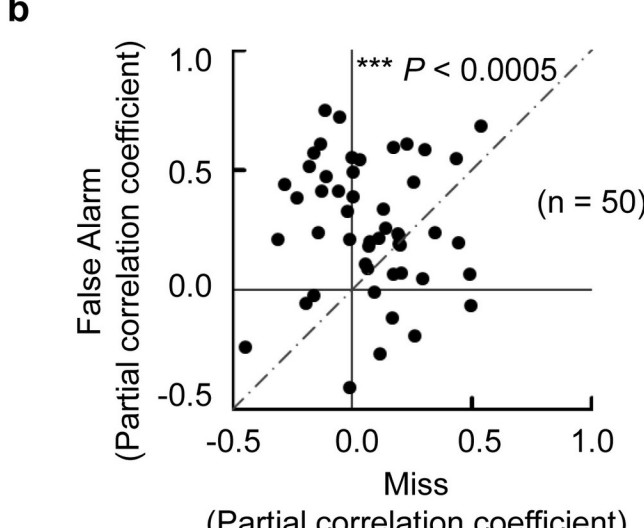

**Fig 7. Neuronal signal predicting animals' behavior. (a)** Example neuron exhibiting target-selective activity. (Left) Raster displays of correct trials sorted by target locations. Colors indicate target locations on display. The neuron exhibited preferred responses when the target location was the top left (red). (Right) Raster displays of error trials and SDFs (σ = 20 milliseconds). False Alarm, top left as the incorrect positions the subjects chose (black). Miss, top left as the correct positions the subjects missed (gray). Correct trials, top left as the correct positions the subjects chose (red). **(b)** Error analysis for target-selective neurons with at least 10 error trials (*n* = 50). False Alarm, the false positions the subjects chose. Miss, the correct positions the subjects missed. Each dot indicates 1 neuron. ***$P < 0.0005$, Wilcoxon's signed-rank test, 2-tailed. Source data are available in S1 Data. SDF, spike density function.

(Fig 7B). These results suggest that the target-selective activity constructed by the HPC neurons predicts the subsequent animal behavior.

## Discussion

The present study aimed to investigate whether the flexible use of past knowledge can be explained by a constructive process in the HPC. We found a robust memory signal reflecting the location information retrieved from an item cue (Fig 2), which was substantial even after the onset of background cue (Fig 3, S5 Fig). The perceptual information of the background cue

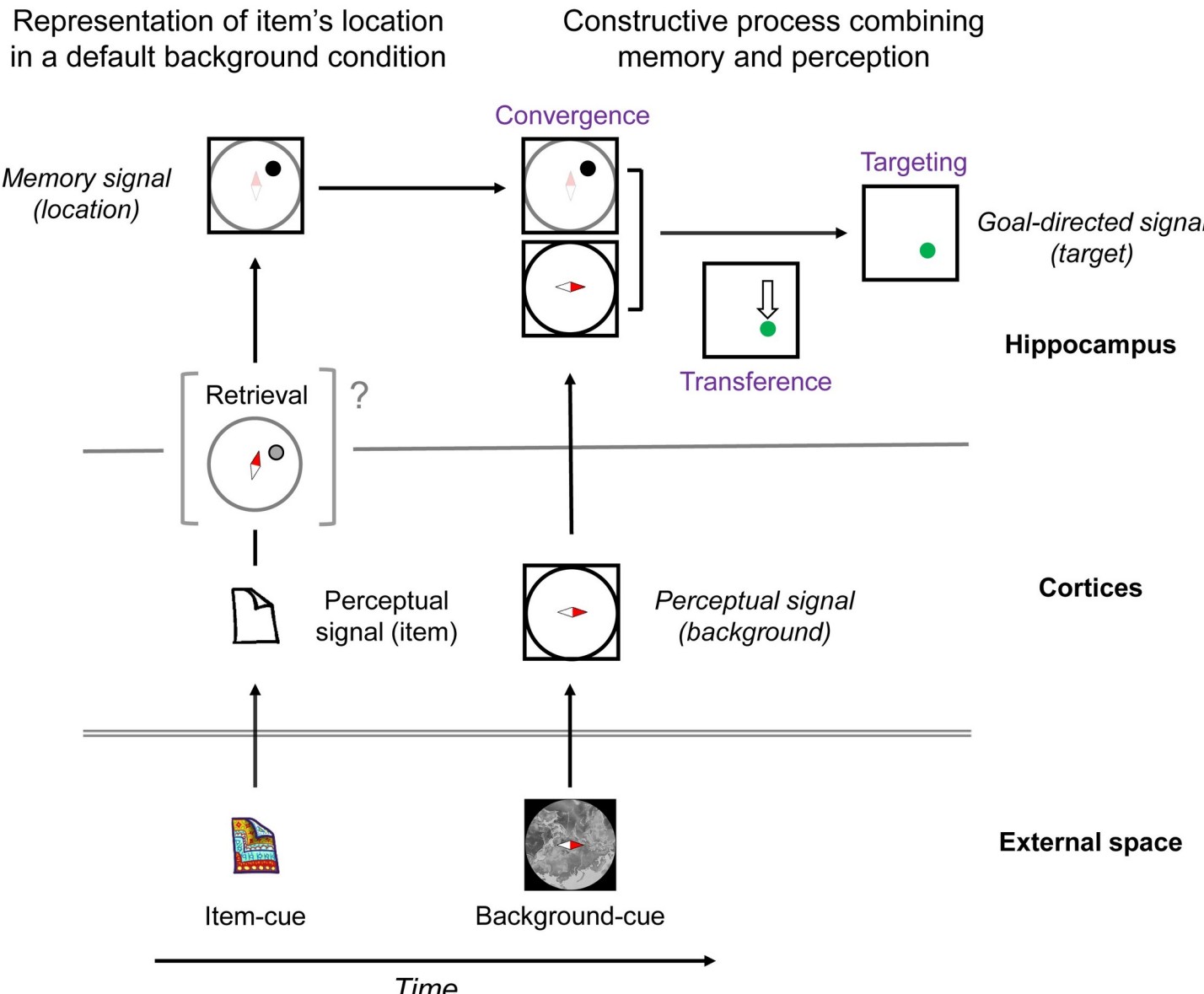

**Fig 8. Constructive process for the flexible use of memory.** Schematic diagram of neuronal signals during a trial of the CMP task, in which the item cue and the orientation of background cue were I-B and 90˚, respectively. In the HPC, the retrieved location of the item is represented relative to the 0˚ background image, which may correspond to the top right in egocentric space. The incoming perceptual signal is integrated with the memory signal to construct an updated information signaling the target location by following sequential neuronal operations: convergence (i.e., memory [co-location I on the 0˚ background] + perception [90˚ background]), transference (i.e., from the top right [co-location I on the 0˚ background] into the bottom right [co-location I on the 90˚ background]), and targeting (i.e., coding bottom right). It is still unknown which brain area is involved first in the retrieval of item-location association memory and whether the retrieved memory content is same as the memory signal in the HPC. CMP, constructive memory-perception; HPC, hippocampus.

was converged on the retrieved location signal (Fig 4), which transferred the retrieved location to the target location (Figs 5 and 6). The target information was correlated with the animal's subsequent behavioral response (Fig 7). The present findings thus indicate that the HPC neurons combine mnemonic information with perceptual information to construct goal-directed representations of the retrieved memory (Fig 8), which would be useful in the current situation for a subsequent action.

In previous electrophysiological studies, the effects of the item-location association memory were presented as learning-dependent changes in firing rates (e.g., changing cells) [8, 26] or selective responses to a particular combination of items and locations [26, 27, 30]. However, these studies did not identify the location signal retrieved from an item cue. In the present study, we evaluated the item-location association memory as correlated responses to the co-location items by assigning 2 visually distinct items to each co-location on the background image (Fig 1). We found that the HPC neurons showed the unitized responses to the co-location items, reflecting the location retrieved from the item cues (Figs 2 and 3). In addition, by orienting the background cue randomly, we dissociated the item-location association memory from the item-response association [3] in the CMP task, which was not clearly dissociated in the previous physiological studies [8, 26, 27]. Together, the CMP task allowed us to examine the correlated responses to the items, which were semantically linked via the location information. The correlated responses to semantically linked items were previously investigated using the "pair-association" task [31–33]. Different from the co-location items linked by the locations in the CMP task, a pair of items was directly associated with each other in the pair-association task. In this item-item association memory paradigm, the memory retrieval signal representing a target item appeared first in the perirhinal cortex (PRC) of the MTL and spread backward to the visual area TE [31, 34, 35]. Future studies should aim to determine whether the location information is retrieved within the HPC [36] or derives from other areas such as the PRC (Fig 8), which has been considered as a core brain region responsible for semantic dementia [37, 38] as well as a hub of converging sensory inputs [34, 39].

In the CMP task, the sequential presentations of the item and background cues temporally separated the memory retrieval from perceiving the environment and allowed us to observe neuronal dynamics that may underlie the constructive process fitting the retrieved memory to the current situation (Fig 8). Sequential presentations of 2 cue stimuli were also applied in previous studies to investigate a conjunctive representation of the 2 stimuli in the HPC of rats [13] and in the PRC of nonhuman primates [40]. In these studies, animals learned to associate combinations of 2 cues with choice responses [13] or reward deliveries [40] directly because the 2 cue stimuli (e.g., sound and odor) did not have any internal relationship and the combinations of 2 cue stimuli were arbitrarily assigned to the correct responses (e.g., pulling right/left lever) or outcomes (e.g., presence/absence of reward). In this condition, the memory retrieval signal appeared after the second cue [13, 40]. Conversely, the combinations of the item and background cues in the CMP task necessarily determined the correct target positions because the item cue was assigned to a particular position on the background-cue image in allocentric coordinate (Fig 1B). To realize this experimental design in an actual animal experiment, we trained the animals on the preliminary stimulus set (S7 Fig) to train the task rule in a relatively easy condition before the main stimulus set (S1 Text, S3 Table). This training procedure would prevent an animal from learning to "solve" a task in a gradual manner like probabilistic learning or inflexible learning of amnesia patients, in which participants learned to "solve" the tasks implicitly, and their performances were supported by the striatum or neocortical areas rather than the hippocampus [41, 42]. The present experimental design, including the training procedure, was validated by the correlated responses of the HPC neurons with the animal's subsequent behavioral response suggesting involvements of the HPC in the CMP.

In addition to the constructive process, the present experimental design revealed the default position/orientation for the background image in the HPC when the animals represented the retrieved location of the item cue in their mind (Fig 8). Considering the training history of the monkeys for the item-location association, the default position may depend on their initial trainings. This finding may explain our mental representations of landmarks for their locations, which depends on our experiences [43]. For example, when you remember locations of

the Statue of Liberty and Brooklyn Bridge in the New York City, you may automatically put them to the bottom and right, respectively, in your mind. An unanswered question in the present study is whether the mental locations were coded in egocentric space relative to the animals' head position [45] or in allocentric space relative to a frame of computer screen for the background image [20, 44] or maybe both [46].

The presence of the default position/orientation for the background image suggested that the constructive process for the goal-directed information was triggered by an update of the background from its default to a current position/orientation during a trial of the CMP task. The HPC neurons carried both past and present information on their activity. This convergent-type activity may lead the targeting-type activity via the transferring-type activity when the background cue was different from the default position/orientation (i.e., "non-match" condition). These sequentially occurring neuronal operations may be useful to construct (cf., retrieve) a target from multiple signals that have an internal relationship (e.g., the retrieved co-location and the background cue in the CMP task). For example, you may answer a direction of the school gymnasium from your current position/orientation in the campus easily even though you do not have a direct experience to go there from your current position [3, 16, 47].

One reasonable question here might be whether the retrieved location was transferred to the target position by a mental rotation [48] of the retrieved location on the background image. If we assume the target information was constructed from the convergent-type of activity including the current background-cue information, which was already oriented, it would be reasonable to consider that the retrieved location was transferred to the target position directly without a transit between the 2 positions in geometric space. The direct transfer may occur only when a subject is familiar with a current environment, in which enough information could be provided to compute the target location accurately in the HPC. Conversely, when the environment is not enough familiar, the subject may recruit an active simulation process like mental rotation, which would be supported by other cortical areas [49–51].

Another unanswered question is whether the target-location-selective activity in the HPC encodes an action plan (i.e., endpoint of the saccade) or a mental representation of a target location itself. In the contextual fear memory paradigm [52], the HPC provides the amygdala with context information rather than its associated valence triggering fear responses [53]. Moreover, recent human functional magnetic resonance imaging (fMRI) studies reported activation of the default mode network during future simulations [54] and suggested that the retrieved information spreads from the MTL to medial prefrontal cortex (PFC) [47], which is reportedly involved in decision-making [55]. On the basis of these findings, we hypothesize that the HPC may provide its downstream regions with a target location, which may guide subsequent action selection [56].

In the present CMP task, we found the 3 neuronal operations in the HPC that were involved in the construction of goal-directed information. Constructive neural processing is best recognized in the visual system [57]. Based on the anatomical hierarchy, the construction proceeds from the retina to HPC through a large number of distinct brain areas to construct a mental image of an entire visual scene from local visual features (e.g., light spots, oriented bars). A recent electrophysiological study demonstrated constructive perceptual processing in the MTL, which combines an object identity with its location when the monkeys look at the visual object [20, 58]. However, a constructive process for perceiving an entire scene is still an unsolved question. As to the memory system, Schacter and his colleagues proposed the "constructive episodic simulation hypothesis" [1, 59], which assumes that our brain recombines distributed memory elements to construct either past episodes or future scenarios (i.e., "mental time travel") [24, 25]. However, the neuronal correlates to the constructive memory process for the mental time travel have not been identified as far as we know. There are preceding

studies reporting neuronal activities signaling both past and future regarding the performance level during a single recording session in the macaque prefrontal cortex [60] or the task events during a single trial in the rat HPC [13], but neither showed the constructive process. In the present study, we exhibited a constructive process in which HPC neurons combined the past knowledge with incoming perception for its flexible use rather than for perception of an entire scene or for mental time travel.

Considering its functional significance as declarative memory, the constructive process operated by the 3 neuronal operations for the flexible use of mnemonic information in the HPC may be shared across species. Moreover, this constructive process combining both memory and perception might be a precedent of the constructive memory process combining only the memory elements for the "mental time travel" in the evolution process of declarative memory system. The underlying neuronal mechanisms of the constructive process in the HPC should be further investigated by theoretical and experimental study across species. The transitions of the 3 neuronal operations for the constructive process might be related with attractor dynamics substantiated by the HPC recurrent networks, which is reportedly involved in spatial memory of rodents [61–64].

## Materials and methods

### Ethics statements

The experiments were performed in accordance with the National Institutes of Health (NIH) Guide for the Care and Use of Laboratory Animals and were approved by the Institutional Animal Care and Use Committee (IACUC) of Peking University (Psych-YujiNaya-1).

### Experimental design

**Subjects.**  The subjects were 2 adult male rhesus monkeys (*Macaca mulatta*; 6.0–9.0 kg).

**Behavioral task.**  We trained 2 monkeys on a CMP task (Fig 1, S1 Text, S7 Fig and S3 Table). During both training and recording sessions, animals performed the task under dim light. The task was initiated by the animal fixating on a white square (0.5˚ visual angle) in the center of a display for 0.5 seconds. Eye position was monitored by an infrared digital camera with a 120-Hz sampling frequency (ETL-200, ISCAN). Then, an item cue (diameter, 3.8˚) and background cue (diameter, 31.4˚) were sequentially presented for 0.3 seconds each with a 0.7-second interval. After an additional 0.7-second delay interval, 4 equally spaced white squares (0.5˚) were presented at the same distance from the center (8.5˚) as choice stimuli. One of the squares was a target, whereas the other 3 were distracters. The target was determined by a combination of the item cue and the background cue stimuli. The animals were required to saccade to one of the 4 squares within 0.5 seconds. If they made the correct choice, 4 to 8 drops of water were given as a reward. When the animals failed to maintain their fixation (typically less than 2˚ from the center) before the presentation of choice stimuli, the trial was terminated without reward. Before the recording session, we trained the animals to associate 2 sets of 4 visual stimuli (item cues) with 4 particular locations relative to the background image that was presented on the tilt with an orientation from −90˚ to 90˚. We first trained the monkeys to learn the task rule of the CMP task using a preliminary stimulus set (monochromatic simple-shaped objects [e.g., cross, heart] as item stimuli and a large disk with 4 monochrome colors in individual quadrants as the background stimulus) (S7 Fig) in the preliminary training before the final training using a main stimulus set (S1 Text). In addition, to avoid that the monkeys learn to associate each combination of the item cue and the background cue with a particular target location, the orientation of background image was randomized at a step of 0.1˚, which increased the number of combinations (8 × 1,800) and would make it difficult for

the animals to learn all the associations among item cues, background cues, and target locations directly (S1 Text). During the recording session, the item cue was pseudorandomly chosen from the 8 well-learned visual items, and orientation of the background cue was pseudorandomly chosen from among 5 orientations (−90˚, −45˚, 0˚, 45˚, and 90˚) in each trial, resulting in 40 (8 × 5) different configuration patterns. We included the trials with −45˚ and 45˚ background cues during the recording session in order to increase the number of configuration patterns and to prevent the animals from linking a combination of the item cue and the background cue to the target location directly, although we did not use these trials in the main analyses. We trained the 2 monkeys using same stimuli but different item-location association patterns. All stimulus images were created using Photoshop (Adobe, https://www.adobe.com/).

**Electrophysiological recording.**    Following initial behavioral training, animals were implanted with a head post and recording chamber under aseptic conditions using isoflurane anesthesia. To record single-unit activity, we used a 16-channel vector array microprobe (V1 X 16-Edge; NeuroNexus) or a single-wire tungsten microelectrode (Alpha Omega), which was advanced into the brain by using a hydraulic Microdrive (MO-97A; Narishige) [11]. The microelectrode was inserted through a stainless steel guide tube positioned in a customized grid system on the recording chamber. Neuronal signals for single units were collected (low-pass, 6 kHz; high-pass, 200 Hz) and digitized (40 kHz) (AlphaLab SnR Stimulation and Recording System, Alpha Omega, https://www.alphaomega-eng.com/). We made no attempt to prescreen isolated neurons. Instead, once we succeeded in isolating any neuron online, we started a new recording session. The offline isolation of single units was performed using Offline Sorter (Plexon, https://plexon.com/) by manual curation to make sure that noise transients were not included as units and that the same cell was not split into several clusters. The cells were isolated depending on the properties of spike waveforms. The cells were included into the analysis if the cells fired throughout the recording session with well-defined fields and a minimal mean firing rate as 1 Hz. On average, 128 trials were tested for each neuron ($n = 456$). The placement of microelectrodes into target areas was guided by individual brain atlases from MRI scans (3T, Siemens). We also constructed individual brain atlases based on the electrophysiological properties around the tip of the electrode (e.g., gray matter, white matter, sulcus, lateral ventricle, and bottom of the brain). The recording sites were estimated by combining the individual MRI atlases and physiological atlases [65].

The recording sites covered between 3 and 16 mm anterior to the interaural line (monkey B, left hemisphere; monkey C, right hemisphere; S2 Fig). The recording sites cover all the subdivisions of the HPC (i.e., dentate gyrus, CA3, CA1, and subicular complex) [11]. A final determination will require future histological verification (both animals are currently still being used).

## Statistical analysis

All neuronal data were analyzed by using MATLAB (MathWorks, https://www.mathworks.com/) with custom written programs, including the statistics toolbox.

**Classification of task-related neurons during the item-cue period.**    We calculated mean firing rates of 8 consecutive 300-millisecond time-bins moving in 100-millisecond steps, covering from 0 to 1,000 milliseconds after item-cue onset in each of all correct trials. We evaluated the effects of "item" for each neuron by using 1-way ANOVA with the 8 item-cue stimuli as a main factor ($P < 0.01$, Bonferroni correction for 8 analysis-time windows). We referred to neurons with significant item effects during any of the 8 analysis-time windows as item-selective neurons. For a comparison of the item-selective activity between the item-cue period and

the background-cue period, we also defined the item-selective neurons during the background-cue period (Fig 3).

**Classification of task-related neurons during the background-cue period.**   We calculated the mean firing rates of 8 consecutive 300-millisecond time-bins moving in 100-millisecond steps, covering from 0 to 1,000 milliseconds after the background-cue onset. We evaluated the effects of "co-location," "background," and "target" for each neuron by using 3-way nested ANOVA with the 4 co-locations, 3 background-cue orientations, and 4 target locations as main factors, and the 8 item-cues nested under the co-locations ($P < 0.01$, Bonferroni correction for 8 analysis-time windows). The 3-way nested ANOVA was conducted using the correct trials with $-90°$, $0°$, and $90°$ background-cues. $-45°$, and $45°$ background-cues were excluded from the ANOVA because they would bring about a bias for the target location (S3 Text). For a comparison of the co-location-selective activity between the item-cue period and the background-cue period, we also defined the co-location-selective neurons during the item-cue period using 3-way ANOVA (S5A Fig). Out of the task-related neurons defined by the 3-way nested ANOVA, we further defined a neuron showing both co-location and background effects ("convergence") during the background-cue period and 2 subcategories of the target-selective neurons ("transference" and "targeting") during the background-cue period (S1 Table).

**Analysis of retrieval signal during item-cue period.**   To show the time course of activity for an individual item-selective neuron, an SDF was calculated using only correct trials and was smoothed using a Gaussian kernel with a sigma of 20 milliseconds.

We examined the retrieval signal of each item-selective neuron by calculating Pearson correlation between the responses to the co-location items {i.e., $[f(\text{I-A}), \ldots, f(\text{IV-A})]$ and $[f(\text{I-B}), \ldots, f(\text{IV-B})]$} (4 pairs, $d.f. = 2$) (S4 Fig). Here, $f(\text{I-A})$ indicates the response to the item I-A. Because individual neurons in the HPC showed various time courses of item-selective activities (e.g., Figs 2A and 3A), we calculated the correlation coefficients during each of the analysis-time windows with a significant item effect for each item-selective neuron ($P < 0.01$, Bonferroni correction for 8 analysis-time bins). We then averaged $Z$-transformed values of the correlation coefficients across the significant analysis-time bins for the neuron. The average value was finally transformed into $r$ value (i.e., co-location index) as shown in Figs 2 and 3.

The retrieval signal was further examined for each item-selective neurons with high co-location index ($r > 0.6$) using the ROC analysis [29, 66]. We calculated a mean firing rate during the item-cue period (60 to 1,000 milliseconds from the item-cue onset) in each trial for the optimal item and its paired co-location items. Cumulative proportions of the trials whose firing rates were larger than a criterion were depicted on the 2-dimensional plot with the optimal item and its paired item as the ordinate and abscissa, respectively. A value of the AUC was evaluated for each neuron by a permutation test. We shuffled the trials for the optimal item and its paired co-location item 10,000 times. At each shuffle, we determined the optimal item and its paired item according to their firing rates and calculated a value of the AUC. Using a distribution of the 10,000 values of the AUC, we determined an expected value (median) and the significance level for each neuron. As a control, we also examined a discrimination between the trials of the best co-locations, including the optimal and its paired co-location items, and the trials of the other co-locations and evaluated a value of the AUC using the permutation test. The ROC analysis was also applied for the item-selective neurons during the background-cue period in the same way.

**Analysis of task-related signal during background-cue period.**   To show the time course of activity for an individual task-related neuron, an SDF was calculated using only correct trials and was smoothed using a Gaussian kernel with a sigma of 20 milliseconds. For comparing time courses of proportions of task-related (co-location, background, and target) neurons and

their signal amplitudes, we conducted 3-way nested ANOVA for each 100-millisecond time-bin moving by 1 millisecond to test significances ($P < 0.01$, uncorrected) with $F$ values for each neuron. The 3-way nested ANOVA was conducted using only the correct trials with −90˚, 0˚, and 90˚ background-cues.

**Analysis of similarity of orientation tuning.** To evaluate the effects of background cue on the co-location-selective responses, we used data in the correct trials with −90˚, 0˚, and 90˚ background cues. As to the example neuron in Fig 4B, we first calculated mean firing rates for each co-location during the 60- to 1,000-millisecond period from an onset of the background cue and determined the "best co-location" and the "second-best co-location" based on the mean firing rates. We then calculated Pearson correlation between responses to the different orientations (−90˚, 0˚, and 90˚) of background cues for the best co-location and those for the second-best co-location (3 pairs, $d.f. = 1$). We also examined a time course of the background-cue effect on the co-location-selective responses by calculating the population-averaged correlation coefficients for each 100-millisecond time-bin moving by 1 millisecond as shown in Fig 4C.

**Calculation of matching index.** To evaluate the relationship between the retrieved location and the target location, we used data in the correct trials with −90˚, 0˚, and 90˚ background cues. For each neuron exhibiting both item-selectivity during item-cue period and target-selectivity during background-cue period, we first averaged responses during the 60- to 1,000-millisecond period from item-cue onset in each trial and calculated a grand mean across trials to each of the 4 co-locations. In addition, we averaged responses during the 60- to 1,000-millisecond period from background-cue onset in each trial and calculated a grand mean across trials to each of the 4 target locations. According to the 3 potential matching patterns, we sorted the firing rates to the co-locations and calculated Pearson correlation coefficients between responses to the co-locations in each of the 3 potential matching patterns {e.g., [$f_{ic}$ (I), $f_{ic}$ (II), $f_{ic}$ (III), $f_{ic}$ (IV)] on 0˚ background cue} and those to the target locations [$f_{bc}$ (TR), $f_{bc}$ (BR), $f_{bc}$ (BL), $f_{bc}$ (TL)] (4 pairs, $d.f. = 2$). Here, $f_{ic}$ (I) indicated an averaged response to the items corresponding to co-location I during the item-cue period, and $f_{bc}$ (TR) indicated an averaged response to the top-right target position during the background-cue period.

**Analysis of target signal.** To evaluate background-cue effect on the target signal, population-averaged SDFs (best−other target locations) were calculated for target-selective neurons across the correct trials with −90˚, 0˚, and 90˚ background cues. We first averaged responses during the 60- to 1,000-millisecond period from background-cue onset in each trial and calculated a grand mean across correct trials to each of the 4 target locations to determine the "best target location" for each neuron. The SDFs to each orientation (−90˚, 0˚, and 90˚) of background cues for all target locations were normalized to the amplitude of the mean response to the best target location, and the normalized SDFs for the best target location was subtracted by the mean normalized responses across the other target locations. The population-averaged SDFs (i.e., target-selective response) were smoothed using a Gaussian kernel with a sigma of 20 milliseconds.

**Error analysis.** We examined whether the target-selective activities signaled positions the subjects chose or correct positions during the background-cue period in error trials by using a partial correlation coefficient. To calculate the partial correlation coefficient for each neuron, we first calculated an average firing rate during each 300-millisecond time-bin moving by 100 milliseconds during the background-cue period (i.e., 8 time-bins in total) for each target position (i.e., 8 positions in total) across the correct trials. We next prepared for 3 arrays for each neuron containing "$n$" elements in each array ("$n$" is the number of error trials for each neuron): (1) firing rates in the *i-th* error trial ($i = 1$ to $n$) (dependent variable, ***D***); (2) the mean firing rate across correct trials with the same target position as the subject chose in the *i-th* error

trial (explanatory variable, *X*); (3) the mean firing rate across correct trials with the same target position as the subject missed (i.e., correct answer) in the *i-th* error trial (explanatory variable, *Y*). The partial correlation coefficients of the dependent variable, *D*, with explanatory variables, *X* and *Y*, were calculated in each time-bin for each neuron when the neuron's responses in correct trials showed a significant target effect ($P < 0.01$, Bonferroni correction for 8 analysis-time windows) and the mean firing rate across trials was larger than 1 Hz in that time-bin. The mean partial correlation coefficients were calculated across the active time bins (i.e., $P < 0.01$, Bonferroni correction for 8 analysis-time windows, >1 Hz) for each neuron using *Z*-transformation.

## Supporting information

**S1 Text. Training procedures for the CMP task.** CMP, constructive memory-perception.
(DOCX)

**S2 Text. Examinations of neuronal responses to co-location stimuli and eye positions.**
(DOCX)

**S3 Text. Detection of task-related signals during the background-cue period.**
(DOCX)

**S1 Table. Numbers of task-related neurons.**
(DOCX)

**S2 Table. Numbers of target-selective neurons selective to each target location.**
(DOCX)

**S3 Table. Numbers of training sessions for the CMP task.** CMP, constructive memory-perception.
(DOCX)

**S1 Fig. Performance in the CMP task.** Performance during recording sessions (*n* = 179 for Monkey B, *n* = 158 for Monkey C). Error bar, standard deviation. Dashed line, chance level = 25%. **(a)** Performance for 8 item stimuli as item cue. Black bars, set A. White bars, set B. **(b)** Performance for 5 orientations of background cue. **(c)** Performance for 8 positions on the display as target locations. Source data are available in S2 Data. B, bottom; BL, bottom left; BR, bottom right; CMP, constructive memory-perception; L, left; R, right; T, top; TL, top left; TR, top right.
(TIF)

**S2 Fig. Recording region.** Magnetic resonance images corresponding to the coronal planes anterior 4 and 10 mm from the interaural line of monkey C (right hemisphere). The recording region is the HPC. A reference electrode implanted in the center of chamber was observed as a vertical line of shadow in the coronal plane at A10. D, dorsal; HPC, hippocampus; L, lateral. M, medial; *ots*, occipital temporal sulcus; V, ventral.
(TIF)

**S3 Fig. Examinations of neuronal responses and eye positions.** Firing rates plotted as a function of eye positions during the item-cue period for the neuron shown in Fig 2A and 2B. Each circle indicates 1 trial. Filled circles indicate trials with the best co-location stimuli as item cues. Open circles indicate trials with the worst co-location stimuli as item cues. The large overlaps were found in the distributions of the eye positions between the trials with the best and worst co-location items (*P* = 0.16 for horizontal, *P* = 0.25 for vertical, *t* test, 2-tailed), whereas distributions of the firing rates were significantly different between the 2 trial types

($P < 0.0001$). These results indicate that the item-selective responses shown in Fig 2 cannot be explained by the animal's eye positions. Source data are available in S2 Data.
(TIF)

**S4 Fig. A schematic illustration of co-location index.** The co-location index was calculated for each neuron as following 5 steps. (1) The item-selectivity was examined in each of 8 consecutive 300-millisecond time-bins using the same threshold (i.e., $P < 0.0125$ for each time-bin, 1-way ANOVA) as that for the definition of item-selective neurons ($P < 0.01$, Bonferroni correction for 8 analysis time-bins). (2) If the time-bin showed a significant item-selectivity, we calculated the correlation coefficient ($r$) between the responses to items from set A and those from set B in the time-bin, and (3) then the $r$ was transformed into $Z$. (4) We then averaged $Z$ values across the significant time-bins and (5) re-transformed the averaged Z value into $\bar{r}$ value as the co-location index of the neuron.
(TIF)

**S5 Fig. Co-location-selective neurons during the item-cue and background-cue periods. (a)** Percentages of neurons showing a significant co-location effect ($P < 0.01$, 3-way nested ANOVA) during the item-cue (I-Cue) and background-cue (B-Cue) periods out of the recorded neurons ($n = 456$). Hatched area, neurons exhibiting co-location-selectivity during both periods. **(b)** Time courses of percentages of neurons showing significant co-location-selective activity and background-selective activity (100-millisecond time bin, $P < 0.01$, 3-way nested ANOVA, uncorrected) out of the recorded neurons (n = 456). Brown bar, presentation of the item-cue. Gray bar, presentation of the background cue. Dashed line, chance level = 1%. Source data are available in S2 Data.
(TIF)

**S6 Fig. Example neurons involved in the constructive process. (a)** Example neuron signaling co-location and background cue in a "convergent" manner. This neuron did not show item-cue selective responses during the item-cue period ($P = 0.34$, 1-way ANOVA), but it exhibited the co-location-selective responses during the background-cue period ($P < 0.01$, 3-way nested ANOVA). The background-selective responses were combined with the co-location-selective responses. The preferred orientation of the background-cue stimulus was 90° for this neuron across co-locations. The same format as Fig 5A. **(b)** Example neuron signaling a "targeting" location. This neuron did not show item-selective responses during the item-cue period ($P = 0.81$), but it exhibited the target-selective responses during the background-cue period ($P < 0.0001$). The best target location was bottom-right of display (green). **(c)** Example neuron that changed the activity patterns, showing multiple operations for the construction (i.e., convergence, transference, and targeting) during the background-cue period. This neuron showed item-selective responses during the item-cue period ($P < 0.0001$), and the preferred items in the item-cue period were I-A, I-B, IV-A, and IV-B. During 300–400 milliseconds after background-cue onset, the background-selective responses were combined with the co-location-selective responses, and the preferred orientation was −90° across co-locations (i.e., "convergence"). During 400–600 milliseconds after background-cue onset, this neuron exhibited strong responses only to the particular combinations of item cue and background cue that corresponded to the top-right target position (blue disk) (blue bars, II-A and II-B, −90° and IV-A and IV-B, 90°) (i.e., "transference"). During 800–1,000 milliseconds after background-cue onset, this neuron exhibited selective responses to the top-right target position regardless of item and background cues (i.e., "targeting"). Source data are available in S2 Data.
(TIF)

**S7 Fig. Stimuli for preliminary training of the CMP task.** (Left) Simple shape objects with monochrome colors as item-cue stimuli. (Right) Large disk with 4 monochrome colors in individual quadrants as a background-cue stimulus. Each item stimulus was assigned to 1 location on the background image. CMP, constructive memory-perception.
(TIF)

**S1 Data. Source data for the main figures.** The source data used to generate main figures are included under the file name "S1_Data.xlsx." Source data for each main figure are arranged by sheet and are labeled. The raw spike files for each neuron are available at https://osf.io/nu9ch/?view_only=1faa4cc2d5254b6eb25740a92e6f693c.
(XLSX)

**S2 Data. Source data for the supplementary figures.** The source data used to generate supplementary figures are included under the file name "S2_Data.xlsx." Source data for each main figure are arranged by sheet and are labeled. The raw spike and eye position files for each neuron are available at https://osf.io/nu9ch/?view_only=1faa4cc2d5254b6eb25740a92e6f693c.
(XLSX)

## Acknowledgments

We thank S. Kitazawa, K. W. Koyano, W. A. Suzuki, I. Lee, K. Miyamoto, S. Fujisawa, H. Chen, and H. Deng for helpful comments and S. Xue for expert animal care. We thank J. Gao, W. Men, G. Yang, and the National Center for Protein Sciences at Peking University for assistance with MRI scanning. We thank D. Lanham for providing the source images of the main stimulus set.

## Author Contributions

**Conceptualization:** Yuji Naya.

**Data curation:** Cen Yang.

**Formal analysis:** Cen Yang.

**Funding acquisition:** Yuji Naya.

**Investigation:** Cen Yang.

**Methodology:** Cen Yang, Yuji Naya.

**Project administration:** Yuji Naya.

**Resources:** Yuji Naya.

**Software:** Cen Yang.

**Supervision:** Yuji Naya.

**Validation:** Yuji Naya.

**Visualization:** Cen Yang.

**Writing – original draft:** Yuji Naya.

**Writing – review & editing:** Cen Yang, Yuji Naya.

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
