## [Editor Report · Decision Letter 0]

22 Apr 2020

Dear Dr Naya, 

Thank you for submitting your manuscript entitled "Hippocampal cells integrate past memory and present perception for the future." for consideration as a Research Article by PLOS Biology.

Your manuscript has now been evaluated by the PLOS Biology editorial staff, as well as by an academic editor with relevant expertise, and I'm writing to let you know that we would like to send your submission out for external peer review. Please accept my apologies for the delay incurred while we sought external advice during these challenging times.

Please re-submit your manuscript within two working days, i.e. by Apr 24 2020 11:59PM.

Kind regards,

Roli Roberts

Senior Editor

PLOS Biology

---

## [Decision Letter · Decision Letter 1]

2 Jun 2020

Dear Dr Naya,

Thank you very much for submitting your manuscript "Hippocampal cells integrate past memory and present perception for the future." for consideration as a Research Article at PLOS Biology. Your manuscript has been evaluated by the PLOS Biology editors, an Academic Editor with relevant expertise, and by four independent reviewers.

You’ll see that the reviewers are broadly positive, but that both reviewers #3 and #4 raise some significant conceptual and analytical issues. There seems to be a common theme that the paper lacks clarity, and would benefit from clearer explanations and presentation of additional (existing) data and some new analyses.

In light of the reviews (below), we are pleased to offer you the opportunity to address the comments from the reviewers in a revised version that we anticipate should not take you very long. We will then assess your revised manuscript and your response to the reviewers' comments and we may consult the reviewers again.

We expect to receive your revised manuscript within 1 month, but do let us know if you require more time, especially under the current circumstances.

**IMPORTANT - SUBMITTING YOUR REVISION**

*Resubmission Checklist*

*Published Peer Review*

*PLOS Data Policy*

*Blot and Gel Data Policy*

Sincerely,

Roli Roberts

Senior Editor

PLOS Biology

REVIEWERS' COMMENTS:

Reviewer #1:

The authors challenged to uncover the neuronal mechanisms underlying the flexibility of declarative memory, i.e., how to flexibly use stored information in changing situations. They recorded single-neuronal activity from the monkey's hippocampus when it was performing a newly developed task, in which retrieval of item-location association memory and flexible use of it was required. They showed that hippocampal neurons were activated by both mnemonic information representing the location and perceptual information representing the external circumstance. They suggest that the two signals were combined at a single-neuron level in the hippocampus and construct goal-directed information by sequentially occurring convergence, transference, and targeting. 

The motivation of this study inspired by "constructive episodic memory system" is sound and challenging. The experimental designs including the new behavioral task (CMP task) in particular are thoughtful and the data analysis is comprehensive, just like the previous excellent studies of Dr. Naya. I have only a few minor comments about discussion, that is, about interpretation of the present data and possible hypothesis for future studies of declarative memory. 

Comment

1. In the CMP task, the "transference" process from the retrieved location into a target location might be a type of "mental rotation", i.e., rotating the retrieved cue location in a mental image to the direction indicated by the background cue. Though the authors already discussed similar issues in lines 258-276, a brief statement discussing the relation of their data and mental rotation in HPC and/or related areas with some references, if possible, may be helpful for the readers. 

2. It might be helpful if the authors would present a summary graph or table of total proportions of the task-related specific neurons, i.e., neurons representing retrieved memory (Fig.2), neurons representing convergence (Fig.3), neurons representing retrieved and target locations (Fig.4), neurons constructing goal-directed information (showing transference effect) (Fig.5), and neurons showing multiple operations (Fig.5). This summary could help us to make an image of information processing of declarative memory in neurons and neuronal population levels in HPC. I expect that the numbers of these neurons are different in a systematic way and more discussion will be possible in future studies. 

3. The authors reported that the signals of mnemonic information and perceptual information were combined at a single-neuron level in HPC. The mechanism of such convergence of different information for past and present to a single neuron is the great finding by the authors. On the other hand, as a conceptual assumption, convergence of different elementary information could be realized by co-activation of neurons related to elementary information alone. It might be helpful for me and the readers if the authors would add a brief discussion of how the mechanism of convergence to individual neurons (convergence neurons) is advantageous to realize the dynamic features of declarative memory. 

Reviewer #2:

In this paper, Yang and Naya investigated hippocampal neuronal activities related to past memory and future goals. They developed a new task that requires the monkeys to associate items and locations in the background rotated trial by trial. They found that hippocampal neurons displayed the information of the item and target locations during the task. They also demonstrate the transitions of neuronal representations from item cue to direction cue presentations. The behavioral and analysis methods are appropriate and persuasive. The findings in this paper are significant and will be of broad interest. I think that the manuscript is suitable for publication in PLoS Biology. I have several comments which may improve the manuscript:

- They classified the target-selective neurons in two types: 'targeting-type' (Fig 4) and 'transferring-type' (Fig 5). The 'targeting-type' neurons were activated when the direction difference between the retrieved location (co-location) and target direction was 0 degree. On the other hand, the 'transferring-type' neurons were activated when the target directions were 90 and -90 degrees from the retrieved location. However, the descriptions of these two types are sometimes confusing in the manuscript. For example, the sentence "these (= item and target selective) neurons tended to show the preferred target locations that corresponded to the preferred co-locations relative to the 0° background-cue (default orientation), but not to the other orientations (Fig 4c)" sounds contradictory because the transferring-type neurons preferred the opposite. (This seems because the analyzed populations for 'transferring-type' and 'targeting-type' are different, but the reason why they selected the different populations for these two types is not clear.) More structural and comprehensive explanations and statistical analysis of these two types of neurons would help readers to understand the importance of the results. Also, adding the numbers of 'targeting-type' and 'transferring-type' neurons in S1 Table would be informative. 

- Fig 3a, b. The spike density functions (top) do not match the results of the F-values (middle) and the firing rates (bottom). For example, SDF shows almost no firing activity in the trials of the item II-AB, but the bar graph shows high firing rates in Fig 3a. It seems that the SDFs of Fig 3a and b were swapped by mistake. 

Reviewer #3:

[IDs himself as Mingsha Zhang]

In the present study, authors trained two monkeys to perform a newly designed task, namely constructive memory-perception task (CMP), in which two visual stimuli were presented sequentially. The crucial point is that the second stimulus was oriented randomly between -90 and 90 degree as the 0 degree being defined as the vertical meridian. Monkeys need to remember the allocentric location of the first stimulus (item-cue) within the second stimulus (background-cue), and determine the goal location of response (saccade) according to the orientation of the background-cue for each individual trial. In this way, authors are able to separate the process of retrieval of remembered item location from the process of perception to current visual stimulus. Then authors explored the neuronal substrate of flexibly using of long-term declarative memory in monkey's hippocampal cortex. They found that hippocampal neurons signaled both mnemonic information, which represented the retrieved item location, and perceptual information, which represented the external circumstance. Thus, authors argue that the hippocampal cortex equips the declarative memory with flexibility in its usage by the constructive process combining memory and perception. While the topic of this study is important and interesting for readers with broad background, there are issues in both conceptual and data analysis that need authors to concern. 

Major comments:

1. Up to my understanding, the long-term declarative memory mentioned in this paper is the spatially-associated memory between item-cue and background-cue. Because two different spatial coordinates were employed in the CMP task, i.e., allocentric coordinate for localizing the item-cue's position within the background-cue; egocentric coordinate for localizing the position of saccadic target, it will be easier for readers to catch up the logic of the task design if authors explicitly describe the exact meaning of item associated location. 

2. A key argument of this paper is that, after the item-cue (I-cue) onset, the activity of HPC neurons represents the retrieved item-location association memory. However, the results of signal neuronal activity (fig 2a) and population neuronal activity (fig 2c) show significant difference between the optimal and pair items, which does not support above argument. Since these two items share the same location, the neuronal response to these items should be similar if it really represents the memorized item associated location. 

3. When looking up the results of fig 2d, there is a subset of hippocampal neurons showing the high correlation coefficient activity between optimal item and its pair item. Suggesting authors to analyze the response of these neurons to the optimal and pair items to see whether the activity is similar. 

4. The CMP task is difficult. To understand the electrophysiology data properly, I would like to suggest authors to present data of monkeys' behavior performance in the results section. 

Minor comments

1. Line 27-29, the words of "south" "north" "turn left" mix the concept of allocentric and egocentric spatial frames. Need to describe these terms clearly.

2. Line 32, it is not clearly defined the meaning of "cognitive map". If I understand correctly, the "cognitive map" here means "allocentric map".

3. Line 41-42, I am not sure the description is correct. Please refer to the paper of Hasegawa et al., 2000 Science.

4. In the caption of fig 2, the subtitle of panel a and c is not suitable because "item-selective neuron" is not consistent with "co-location" neuron in the text. 

5. Line 112-113, authors wrote "This neuron showed the strongest response across all the co-location when the orientation of the background-cue was 90, while........". However, in fig 3a it seems that, in II-A/B, the data of the background-cue with 90-degree orientation is not significant higher than background-cue with 0-degree orientation. 

6. In fig 3, the raster plot of panel a and panel b might be miss-displaced, please have a check.

7. In fig 3, 4 and 5, the term "I-cue" and "B-cue" were not defined in both text and figure caption. 

In the results section of text, there is no description of what data are represented by the three curves (yellow, grey and black) in fig 3a-b, as well as in fig 4a and fig 5a-b. 

8. Line 162, under this subtitle authors show neurons responded to specific target location in fig 5. Are there neurons response to 4 target locations, respectively? 

9. Authors need to explicitly discuss the advantage of the present study (task design, etc.) in study of memory flexibility, comparing with previous studies.

Reviewer #4:

The manuscript by Yang & Naya seeks to examine how the hippocampus integrates retrieved information and current perceived information for subsequent decision making. The main strength of the paper is the multi-step task used, which allows the authors to examine different phases in which memory is first retrieved, a subsequent stimulus is presented, and a response is made. Since the response depends on both retrieval and perception, this requires integration of memory and perception. The authors present different flavors of hippocampal neurons which show varied selectivity profiles, showing both neurons that combine retrieved and perceptual information, along with those that show responses related to the choice. The results expand the view of different types of hippocampal neurons and how single neurons integrate a variety of sources of information during a complex task. The link between neural responses and behavior on error trials also reveals that hippocampal signaling is predictive of future behavior rather than accuracy per se. However, I think that several details need to be clarified, and more broadly the manuscript could be improved by preparing the reader for the types of neurons being examined. 

- The authors frame the paper around the use of memory in a flexible manner, and how this occurs at a single neuron level. The notion of "convergence" or neurons demonstrating selectivity based on a combination or conjunction of cues is commonly queried in hippocampal neurons, so this property was expected (although, it could be motivated based on prior literature in the Introduction, this literature is considered only in the Discussion). However, I found the notion and use of "transference" to be confusing/unclear and not well-motivated or explained in the Introduction. This made it difficult to interpret this property when presented in the Results. It seems like this response property of neurons emerged as an artifact of how the animals were trained and therefore may not be a general feature of how the hippocampus flexibly integrates information; it likely reflects a form of rotation performed during this task. It would be helpful to explain this up front in the manuscript, given that substantial space is devoted to it.

- Related to the above point, I wonder how general the authors think their findings are (the demonstration of 3 main types of neurons showing 'convergence', 'transference', and 'targeting'), or whether the specific types of neurons found here are strongly tied to the specific task and the amount of time needed to train animals to perform this kind of task. For example, is there an over-representation of 'transference' neurons simply due to the considerable training on the task and the manner in which the animals were trained (which is already evident in the 'transference' neurons), or is this a general feature of an episodic memory system? It is clear that the authors favor the later interpretation (third paragraph of the discussion), but a broader discussion of whether all of the response properties seen here are related to episodic memory versus considerable training on a specific behavior task should be addressed. 

- Several correlation analyses are presented in the manuscript (Figure 2, co-location indices, Fig. 3d, similarity of background tuning, and in Fig. 4). Although I understood the theoretical significance of the 'co-location index' (Fig. 2), I generally found these correlation measures to be confusing, because 1) the description of these analyses, i.e. the data going into each of the correlations was not clearly stated and 2) the theoretical significance of each of the different correlations could be much more clearly spelled out in the Results (at least in my reading for those shown in Figs. 3 and 4). Both should be clarified - otherwise these analyses could be completely lost on the reader (I had to expend considerable effort to try to figure out what was being described/shown). For example, it was not clear to me how the correlation for the co-location index shown in 2b (example neuron) and 2d (population) was computed. In the description of the analysis in the Methods (lines 377-382), the sets of data points that are being correlated with each other are not actually stated. Is this the correlation between the mean firing rate in response to items within a co-location pair, such as I-A and I-B, across individual time bins (the 8 relevant time bins) or across individual trials and time bins? Without knowing what the correlation is computed over (i.e. the correlation in response across trials, or the correlation over time bins of mean response), the reader has no sense of the data going into the correlation, so it makes it difficult to interpret. I inferred that the correlation is being computed across time bins using the mean firing rate over trials, between the two co-location stimuli. This is potentially concerning if there are a low number of data points (are all 8 time bins included, or just those with significant F-stats?). Again, this can be addressed by clearly describing the analysis procedure and indicating what the correlation is being computed between (lines 377-382, 397-398, 409-411). Similarly, the data points going into the correlations shown in Figs. 3b and 3d were not clear to me. Is this the correlation between mean responses across the 3 background orientations for the 'best' and 'second-best' co-location stimuli, such that three data points are going into the correlation? Note that additional schematics may be helpful in illustrating these analyses.

Minor comments:

- The training performed prior to the recording sessions is described in the SI text. It is easy for the reader to miss this description - please reference this text in the Methods where training is mentioned (lines 318-320). 

- Please include a few more details regarding the training. It looks like there are several phases of training: training on 4 stimuli with a 0 degree background cue, training on 4 stimuli with varied background cues (from -90 to 90), and then training on the 8 stimuli which were actually used in the recording sessions. Please describe the approximate duration of training for each of these phases prior to the recording task- how long did it take for the animals to be trained in each of these phases? In the final part of training, when the animals were trained on the 8 stimuli used in the paper, were they again trained first with the 0 degree background cue and then introduced to the varying direction of the cue? Were they trained on all 8 stimuli at once? Please include these details.

- Three-way ANOVAs were performed across sliding windows to examine the effects of 'co-location', 'background' and 'target'. Yet only 3 of the 5 backgrounds were used in this analysis (-45 and 45 degrees were excluded). What is the rationale for removing 2/5 of the data from the analysis? I found this to be particularly confusing when trying to understand the data shown in Figs. 3-5, since the top panels show firing rates over time for all 5 background directions, yet when the data are broken down by background direction in the panels below, only 3 directions are shown. 

- On lines 127-129 it is stated that "After background-cue presentation…neurons…exhibited either co-location-selective activities (14%) or background-selective activities (14%)." Yet, these numbers conflict with the data shown in Fig. 3C - with ~7% of neurons showing each of these properties after the background cue is presented. 

- To show 'convergence' of retrieved information with the background cue ('perceptual') information, the authors analyze orientation tuning for co-location selective neurons (in Fig. 3D, 66 neurons). Were the co-location-selective neurons (those showing a 'retrieval' signal) defined only from the time period after the item cue was presented (before the background cue was shown)? Please clarify - I could not find this in the Methods. This definition would provide the clearest evidence that neurons which initially signal retrieval also show selectivity to the background perceptual information. 

- Related to the above point, i.e. demonstrating 'convergent' retrieval signals and perceptual ('background') signals within individual neurons, I wonder if similar results are obtained by simply looking at the intersection of neurons which show both co-location selective information and background-selective information (intersection of neurons shown the individual effects presented in Fig. 3D). If the proportion of neurons demonstrating both kinds of selectivity are greater than would be expected by chance, this is another way to demonstrate the presence of convergent information at the single neuron level (one main point of the paper).

---

## [Decision Letter · Decision Letter 2]

6 Aug 2020

Dear Dr Naya,

Thank you for submitting your revised Research Article entitled "Hippocampal cells integrate past memory and present perception for the future." for publication in PLOS Biology. I have now obtained advice from two of the original reviewers and have discussed their comments with the Academic Editor. 

Based on the reviews, we will probably accept this manuscript for publication, assuming that you will modify the manuscript to address the remaining points raised by reviewer #4. Please also make sure to address the Data and other policy-related requests noted at the end of this email.

We expect to receive your revised manuscript within two weeks. Your revisions should address the specific points made by each reviewer. In addition to the remaining revisions and before we will be able to formally accept your manuscript and consider it "in press", we also need to ensure that your article conforms to our guidelines. A member of our team will be in touch shortly with a set of requests. As we can't proceed until these requirements are met, your swift response will help prevent delays to publication.

*Copyediting*

*Published Peer Review History*

*Early Version*

*Submitting Your Revision*

Sincerely,

Roli Roberts

Senior Editor,

rroberts@plos.org,

PLOS Biology

ETHICS STATEMENT:

-- Please include the full name of the IACUC/ethics committee that reviewed and approved the animal care and use protocol/permit/project license. Please also include an approval number.

DATA POLICY:

Many thanks for providing the raw data in your OSF deposition. However, we also ask that all individual quantitative observations that underlie the data summarized in the figures and results of your paper be made available in one of the following forms:

Regardless of the method selected, please ensure that you provide the individual numerical values that underlie the summary data displayed in the following figure panels as they are essential for readers to assess your analysis and to reproduce it: Figs 2BDG, 3BD, 4AB, 5AC, 6AB, 7AB, S1ABC, S3, S5A, S6ABC. NOTE: the numerical data provided should include all replicates AND the way in which the plotted mean and errors were derived (it should not present only the mean/average values).

REVIEWERS' COMMENTS:

Reviewer #3:

[identified himself as Mingsha Zhang]

Authors properly answered my questions and made modification in the manuscript accordingly. I don't have further questions and comments. 

Reviewer #4:

[identifies herself as Arielle Tambini]

The manuscript has substantially improved with the revision, the authors have thoroughly addressed the points raised. The inclusion of schematics depicting the analysis approaches is much appreciated and I think will benefit the manuscript. I just have a few remaining comments which are minor suggestions. 

Line 143-4: "One critical concern here might be whether the association effect reflected the locations retrieved from the item-cues." The implication or tone of this sentence (that it is problematic or concerning that the co-location effect may reflect retrieval processes) does not match the rest of the paragraph - which does show that the co-location signal is most consistent with reflecting retrieval of a common location. It may be helpful to revise this sentence.

The schematic in the new Fig S7 is a nice addition that helps to clarify the examined memory signals and processes in this task. I'd suggest moving it to the main text/combining it with a figure in the main text so that more readers may see it.

---

## [Editor Report · Decision Letter 3]

22 Sep 2020

*Dear Dr Naya,

On behalf of my colleagues and the Academic Editor, Jozsef Csicsvari, I am pleased to inform you that we will be delighted to publish your Research Article in PLOS Biology. 

Early Version

PRESS 

Kind regards,

Alice Musson

Publishing Editor, 

PLOS Biology

on behalf of

Roland Roberts,

Senior Editor

PLOS Biology